# HybridFlow: Quantification of Aleatoric and Epistemic Uncertainty with a Single Hybrid Model

**Peter Van Katwyk**                                                    *peter_van_katwyk@brown.edu*
*Department of Earth, Environment, and Planetary Sciences, Brown University*
*Data Science Institute, Brown University*

**Karianne J. Bergen**                                                  *karianne_bergen@brown.edu*
*Department of Earth, Environment, and Planetary Sciences, Brown University*
*Data Science Institute, Brown University*
*Department of Computer Science, Brown University*

**Reviewed on OpenReview:** *https://openreview.net/forum?id=xRiEdSyVjY*

## Abstract

Uncertainty quantification is critical for ensuring robustness in high-stakes machine learning applications. We introduce HybridFlow, a modular hybrid architecture that unifies the modeling of aleatoric and epistemic uncertainty by combining a Conditional Masked Autoregressive normalizing flow for estimating aleatoric uncertainty with a flexible probabilistic predictor for epistemic uncertainty. The framework supports integration with any probabilistic model class, allowing users to easily adapt HybridFlow to existing architectures without sacrificing predictive performance. HybridFlow improves upon previous uncertainty quantification frameworks across a range of regression tasks, such as depth estimation, a collection of regression benchmarks, and a scientific case study of ice sheet emulation. We also provide empirical results of the quantified uncertainty, showing that the uncertainty quantified by HybridFlow is calibrated and better aligns with model error than existing methods for quantifying aleatoric and epistemic uncertainty. HybridFlow addresses a key challenge in Bayesian deep learning, unifying aleatoric and epistemic uncertainty modeling in a single robust framework.

## 1 Introduction

Uncertainty quantification plays a critical role in modern machine learning, particularly in high-stakes applications such as autonomous driving (Grewal et al., 2024), medical diagnosis (Jalal et al., 2024), and scientific modeling (Wang et al., 2023). The ability to estimate uncertainty allows models not only to make predictions but also to provide insights into the reliability of those predictions. This capability is especially valuable in scenarios where decision-making depends heavily on understanding the limitations and confidence of a model's outputs (Thuy & Benoit, 2024; Marusich et al., 2024; Amodei et al., 2016). In recent years, researchers have increasingly focused on improving uncertainty estimation methods in deep learning to achieve more robust and trustworthy systems (Gal & Ghahramani, 2016; Kendall & Gal, 2017; Messuti et al., 2025; Yoon & Kim, 2024; Hwang & Shin, 2024). Despite significant progress, key challenges remain, particularly in achieving accurate predictions while effectively quantifying different types of uncertainties (Psaros et al., 2023).

In the context of machine learning, uncertainty is broadly categorized into two types: *aleatoric uncertainty*, which arises from the inherent noise and variability in data, and *epistemic uncertainty*, which reflects the model's lack of knowledge, often due to limited or biased training data (Der Kiureghian & Ditlevsen, 2009; Gruber et al., 2023). While aleatoric uncertainty is irreducible, epistemic uncertainty can, in principle, be reduced by acquiring additional data or improving the model architecture. Techniques such as Monte Carlo

(MC) Dropout (Gal & Ghahramani, 2016) and ensemble methods (Lakshminarayanan et al., 2017) have been developed to estimate epistemic uncertainty, while approaches like density estimation (Bishop, 1994) and heteroscedastic regression (Kendall & Gal, 2017) aim to model aleatoric uncertainty. Despite advancements in single-model frameworks, they often prioritize state-of-the-art uncertainty quantification at the expense of modularity and flexibility, which are critical for integrating uncertainty estimation into diverse existing architectures. For example, methods such as heteroscedastic regression (Kendall & Gal, 2017) are widely adopted due to their simplicity and ease of integration, even though they often suffer from calibration issues and suboptimal predictive accuracy (Seitzer et al., 2022). Heteroscedastic regression methods remain popular because they can be applied to diverse predictive models without significant architectural changes.

This paper introduces a novel hybrid flow-based architecture, HybridFlow, which addresses these challenges by quantifying both aleatoric and epistemic uncertainty without compromising predictive performance. This framework integrates a normalizing flow (NF) (Rezende & Mohamed, 2015) to model aleatoric uncertainty with any probabilistic predictor model for epistemic uncertainty estimation and model prediction. Hybrid-Flow enables precise and reliable uncertainty quantification through leveraging both input data and the latent space generated by the NF as model inputs. By decoupling the aleatoric uncertainty quantification mechanism from the predictor model itself, the proposed HybridFlow framework is inherently modular, serving as a flexible method that can be adapted to a wide range of applications. While the current implementation demonstrates its efficacy using specific benchmarks and case studies, the design of the framework allows for the substitution of different predictors tailored to other tasks or domains.

We evaluate the HybridFlow model framework against three widely adopted frameworks (Kendall & Gal, 2017; Seitzer et al., 2022; Caprio et al., 2024) for quantifying both aleatoric and epistemic uncertainty in a single model, and demonstrate that our proposed framework provides accurate predictions and quantified uncertainty for a range of tasks: depth estimation in computer vision and a series of regression benchmarks. We also include a scientific case study, where we test the HybridFlow architecture as an emulator for future sea level rise. Another key contribution of this work is that we provide a comprehensive evaluation of quantified uncertainty, which is often overshadowed by the analysis of prediction accuracy alone (Wang et al., 2025). We employ a diverse set of metrics that collectively assess different aspects of uncertainty estimation to compare the quantified uncertainty from various methods. The proposed framework showcases a direct improvement in both predictive accuracy and uncertainty quantification over existing methods, providing a compelling step toward single, flexible models that can effectively model both aleatoric and epistemic uncertainties while maintaining high predictive accuracy.

## 2 Related Work

Uncertainty quantification has received significant attention in Bayesian deep learning due to its importance in improving model reliability and interpretability. Uncertainty quantification techniques can be model-agnostic, meaning that they can be incorporated into a variety of predictive architectures, or model-specific, where specific constraints are placed on the predictive architecture in order to quantify uncertainty. Kendall & Gal (2017) laid the foundation for model-agnostic uncertainty quantification by modeling both aleatoric and epistemic uncertainty within a single framework, specifically in the context of computer vision, by using a combination of existing Bayesian approaches and a heteroscedastic log likelihood loss for modeling aleatoric uncertainty. Their approach adds a prediction layer and variance layer at the end of a predictor model, such that the model predicts both the mean $\mu(x)$ and variance $\sigma^2(x)$ for a given input $x$, with the assumption that the uncertainty follows a Gaussian distribution. To find the optimal weights, maximum likelihood estimation is used, which is equivalent to minimizing the negative log-likelihood (NLL), or heteroscedastic loss, of the predictive distribution:

$$\mathcal{L} = \frac{1}{N} \sum_{i=1}^{N} \frac{1}{2\sigma^2(\mathbf{x}_i)} \|y_i - \mu(\mathbf{x}_i)\|^2 + \frac{1}{2} \log \sigma^2(\mathbf{x}_i) \tag{1}$$

For the predictor model, Kendall & Gal (2017) integrate MC Dropout (Gal & Ghahramani, 2016) to estimate epistemic uncertainty, and demonstrate the effectiveness of heteroscedastic regression in tasks such as semantic segmentation and depth regression. This approach enables flexibility, allowing for uncertainty quantification with minimal adjustment to existing predictor model architectures (Smith et al., 2024). How-

ever, this method is prone to calibration issues, particularly under distributional shifts or when noise patterns deviate from Gaussian assumptions. Additionally, reliance on a single combined loss function often leads to overestimation of one type of uncertainty at the expense of the other.

Recent research in model-agnostic uncertainty quantification techniques has highlighted the challenges and pitfalls of heteroscedastic uncertainty estimation. Specifically, Seitzer et al. (2022) critiqued this standard approach to modeling aleatoric uncertainty, emphasizing the tendency of NLL-trained probabilistic neural networks to underestimate uncertainties in the presence of out-of-distribution data or highly skewed noise distributions. They show that the use of the NLL loss is appropriate if the difference between the model prediction and the true value is solely caused by noise, or aleatoric uncertainty. However, in practice, the inability of the model to perfectly approximate the phenomena may also contribute to error, which leads to an overestimation of aleatoric uncertainty when the data is poorly predicted by the model (Seitzer et al., 2022). To address this, they introduce the Beta-NLL loss, a refinement of the NLL loss that adjusts the balance between MSE and uncertainty estimates to address sub-optimal fits and training instability. This method represents an effort to mitigate calibration challenges while retaining the simplicity of the heteroscedastic regression framework. While performance may improve in some tasks, the method remains unstable and fails to resolve miscalibration caused by joint loss functions. Others have attempted to address miscalibration and the decrease in predictive accuracy using differing probabilistic predictors or post-hoc methods, but they similarly encounter comparable trade-offs between predictive accuracy and uncertainty quantification capabilities (Valdenegro-Toro & Mori, 2022; Yang & Li, 2023).

These limitations have motivated a shift toward more expressive generative approaches for modeling uncertainty, particularly those that can capture complex, non-Gaussian output distributions. Normalizing flows (Rezende & Mohamed, 2015) have frequently been used to estimate aleatoric uncertainty by directly estimating the conditional output distribution (e.g., Acharya et al. (2025); Zhang et al. (2024a); Ancha et al. (2024)). To model both aleatoric and epistemic uncertainty using NFs, methods such as NFlows Base and NFlows Out (Berry & Meger, 2023) and FlowNet (Zhang et al., 2024b) use NFs to capture both aleatoric uncertainty (from the learned distribution) and epistemic uncertainty (via ensembling). While these methods demonstrate the expressive capabilities of NFs for modeling uncertainty, they either focus solely on aleatoric uncertainty, or require the flow itself to serve as the predictive model, making the approach inherently model-specific and limiting integration into existing predictors. In contrast, HybridFlow uses the normalizing flow as a density estimator to model aleatoric uncertainty, while allowing prediction and epistemic uncertainty quantification to be estimated using a probabilistic predictor. To our knowledge, HybridFlow is the first to decouple aleatoric and epistemic uncertainty in this way using conditional flows, while retaining architectural modularity and general-purpose applicability. This decoupled design offers greater flexibility and compatibility across applications, particularly where domain-specific predictors are already better suited for prediction tasks.

Other model-specific approaches demonstrate similar tradeoffs between flexibility and performance. For example, Credal Bayesian Deep Learning (CBDL) (Caprio et al., 2024) uses a credal set of Bayesian neural networks (BNNs), with a prescribed set of priors to estimate aleatoric and epistemic uncertainty. CBDL often achieves strong calibration by averaging over the credal set of network, but this approach entails significant computational overhead and requires specialized inference protocols that can be difficult to integrate into existing pipelines. Similarly, Evidential deep learning methods (Sensoy et al., 2018) learn the parameters of higher-order distributions directly from the data, but require specialized architectures and loss functions that restrict their usability. Chan et al. (2024) introduce HyperDM, which employs a Bayesian hyper-network to generate an ensemble of model weights and combines it with a conditional diffusion model to estimate predictive distributions. These methods achieve state-of-the-art uncertainty quantification but require specific model architectures, limiting their practicality for users aiming to enhance existing models or create models for specific tasks and domains where the predictor proposed in these studies may be unfit.

Unlike the aforementioned methods, which integrate uncertainty quantification deeply into the custom architectures, methods such as our proposed HybridFlow framework and heteroscedastic regression (Kendall & Gal, 2017; Seitzer et al., 2022) are flexible by design, allowing integration of uncertainty quantification into any type of probabilistic predictive model. The importance of this capability is evident in the wide-spread adoption of heteroscedastic regression methods (Roohani et al., 2024; Huang et al., 2023; Yelleni et al., 2024),

over custom models with complex, nonflexible model architectures. In this work, we introduce HybridFlow, a hybrid model architecture that surpasses prior flexible heteroscedastic regression frameworks by providing accurate, calibrated prediction and uncertainty estimates while avoiding combined loss function training and generalizing beyond Gaussian assumptions. This enables users to incorporate uncertainty modeling without being tied to specific predictor architectures or objectives. This approach aligns with the practical needs of many users who primarily aim to enhance existing predictors with uncertainty quantification, rather than adopting entirely new architectures that are not specific to applications or domains. HybridFlow is able to provide a balance between advanced uncertainty quantification and compatibility with diverse predictive frameworks.

HybridFlow remains modular and flexible by using a hybrid model architecture, as introduced in Nalisnick et al. (2019). Nalisnick et al. (2019) integrate a generative model with a predictive model, enabling exact computation of the joint distribution $p(x, y)$. Their work demonstrates a theoretically principled integration of tasks but does not explicitly disentangle aleatoric and epistemic uncertainty, and relies on a non-probabilistic generalized linear model for prediction, which limits the expressiveness of the predictive component. HybridFlow builds on Nalisnick et al. (2019) by explicitly using the generative component to model aleatoric uncertainty and by introducing flexibility in the choice of predictive architectures.

## 3    Method

HybridFlow builds upon recent advancements in uncertainty quantification and generative modeling to address existing challenges in modeling both aleatoric and epistemic uncertainty in a single model. It leverages NFs (Rezende & Mohamed, 2015; Papamakarios et al., 2017) for modeling aleatoric uncertainty while integrating a probabilistic prediction model into a hybrid architecture (Nalisnick et al., 2019) for both accurate predictions and epistemic uncertainty quantification. This section describes the components of HybridFlow and their integration into a unified framework for uncertainty quantification.

### 3.1    Conditional Normalizing Flows for Aleatoric Uncertainty

Conditional Normalizing Flows are a class of generative models that transform a simple base distribution $p(z|x)$, typically a conditional Gaussian (Bishop, 2006), into a more complex target distribution $p(y|x)$ through a series of invertible transformations $f$ (Papamakarios et al., 2017; 2021). In the case of conditional NFs, the transformation is parameterized based on the conditioning variable $x$:

$$y = f(z|x), \quad z \sim p(z|x), \tag{2}$$

where $f$ is designed to be invertible, ensuring that the probability density function of $y$ given $x$ can be computed via the change-of-variables formula:

$$p(y|x) = p_z(f^{-1}(y|x)) \left| \det \frac{\partial f^{-1}(y|x)}{\partial y} \right|, \tag{3}$$

where $\frac{\partial f^{-1}(y|x)}{\partial y}$ represents the Jacobian of the transformation $f^{-1}$ conditioned on $x$. The Jacobian matrix captures how the volume of the transformed space changes under the inverse mapping $f^{-1}$. The absolute value of its determinant accounts for this local volume change and ensures the resulting density $p(y|x)$ remains normalized.

In HybridFlow, a *Conditional Masked Autoregressive Flow (CMAF)* (Papamakarios et al., 2017) is used. The CMAF decomposes the conditional joint distribution $p(y|x)$ into a product of autoregressive conditionals:

$$p(y|x) = \prod_{i=1}^{d} p(y_i|y_{<i}, x), \tag{4}$$

where $d$ is the dimensionality of $y$, and each conditional $p(y_i|y_{<i}, x)$ is parameterized by a neural network. This conditional structure allows the flow to explicitly learn the distribution of $y$ given $x$, capturing both the inherent noise in $y$ and its dependence on $x$. Learning $p(y \,|\, x)$ allows for the calculation of aleatoric

uncertainty (Section 3.3), and the invertibility of the flow $f$ enables us to produce a latent representation of the target $y$ conditioned on the inputs, $z = f^{-1}(y|x)$, which serves as an additional feature for the predictor model (Section 3.2). During training, for a given ground-truth pair $(x, y)$, $z$ is deterministically computed using the inverse of the flow. At inference time, however, the ground-truth $y$ is unknown, so z is calculated by taking the expectation over the learned conditional base distribution $p(z|x)$ from the flow. We use a CMAF, as opposed to other NF architectures, due to their strong performance on conditional density estimation tasks, efficient likelihood computation, stable training dynamics, and wide adoption (Papamakarios et al., 2021). However, other flow types (e.g., c-GLOW (Lu & Huang, 2020), RealNVP Dinh et al. (2017)) can also be used in HybridFlow when parameterized conditionally. The framework is agnostic to the specific flow architecture, and different flows may be preferable depending on task complexity, output dimensionality, or computational budget.

While normalizing flows are highly effective generative models for capturing complex conditional distributions, they are not typically used as predictors (Kobyzev et al., 2020). Their primary objective is to model the full data distribution $p(y|x)$ rather than to optimize predictive accuracy for a specific target variable. In contrast, predictive models are explicitly trained to minimize prediction error and can incorporate mechanisms for estimating epistemic uncertainty. By introducing a dedicated predictor, HybridFlow leverages the strengths of both components: the NF captures the uncertainty inherent in the data, while the predictor focuses on producing accurate point estimates and quantifying model-based uncertainty.

## 3.2 Hybrid Architecture

To achieve both predictive accuracy and robust uncertainty quantification, the *latent space* representation $z$ generated by the conditional NF is passed, along with the original inputs $x$, into a probabilistic predictor, as seen in Figure 1. In practice, $x$ and $z$ are concatenated at the feature level; for tabular datasets we directly concatenate the raw input vector with $z$, and for high-dimensional inputs (such as images), a feature extractor (e.g., a CNN with average pooling) is used to learn a compressed feature representation $\phi(x)$ that is concatenated to $z$ before being inputted to the predictor (for more details, see Appendix A)

The probabilistic predictor models the distribution $p(y \mid z, x)$, where $y$ represents the prediction target. By combining $z$ (the latent encoding capturing data-specific uncertainty) and $x$ (the original inputs), the predictor jointly learns the relationship between the inputs and the outputs while retaining uncertainty information from the NF. The design of HybridFlow is flexible, such that any probabilistic predictor can be used. For example, methods such as MC Dropout (Gal & Ghahramani, 2016), Bayesian neural networks, deep ensembles (Lakshminarayanan et al., 2017), and other Bayesian methods are all viable predictor models that can be implemented within the hybrid architecture.

In the HybridFlow framework, the predictive mean $\mu(x, z) = \mathbb{E}[y \mid x, z]$ represents the model's prediction, while the epistemic uncertainty $\sigma_{\text{ep}}^2(x, z) = \text{Var}[y|z, x]$ quantifies the uncertainty due to the model's limited knowledge. The choice of the probabilistic predictor can be tailored based on the application's requirements for computational efficiency, model complexity, or interpretability.

## 3.3 Aleatoric Uncertainty Estimation

The aleatoric uncertainty is captured directly from the learned data distribution modeled by the conditional NF. By sampling from the posterior distribution $p(y \mid x)$ learned by the CMAF, we define the sampled predictions as $\tilde{y} \sim p(y \mid x)$. The aleatoric uncertainty is then computed as the variance of these samples, $\sigma_{\text{al}}^2(x) = \text{Var}(\tilde{y})$.

This variance can be considered aleatoric because it arises from the distribution $p(y|x)$ that the flow explicitly learns from the data. The NF is trained to model the full conditional distribution of possible outcomes given each input, including any inherent noise or ambiguity in the data. When we sample multiple outputs for the same input, the spread of those outputs reflects the learned variability within the data itself. Thus, the variance of these samples naturally corresponds to aleatoric uncertainty, as it captures the irreducible variability in outcomes.

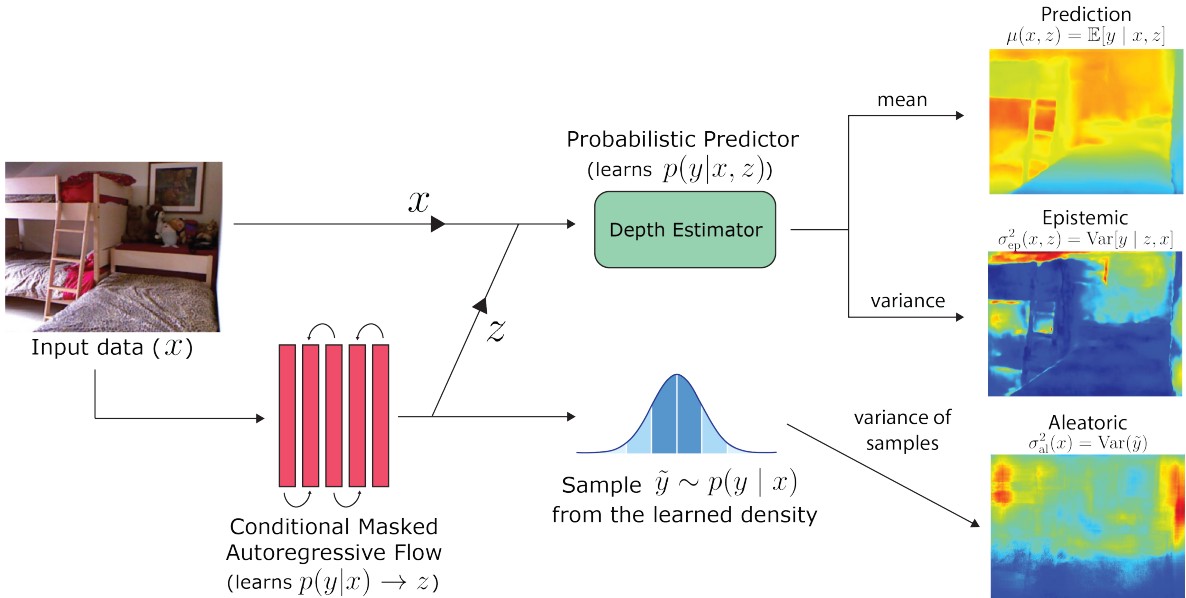

Figure 1: HybridFlow architecture, with depth estimation with the NYU Depth v2 dataset as the example task. A Conditional Masked Autoregressive Flow (CMAF) learns the density of input data (x) through a latent representation (z). The latent representation and the input data are concatenated and used as inputs to a probabilistic predictor model. From the predictor model, the prediction and the epistemic uncertainty can be estimated. The aleatoric uncertainty is estimated by calculating the variance of samples generated from the learned data distribution by the CMAF.

### 3.4 Training Protocol and Implementation Considerations

A summary of the HybridFlow workflow is as follows:

1. **Data Transformation**: The CMAF-based NF, conditioned on inputs $x$, is used to transform the inputs into a latent representation $z$.
2. **Prediction and Epistemic Uncertainty**: The latent $z$ and input $x$ are fed into a probabilistic predictor that outputs the mean prediction $\mu(x, z)$ and epistemic uncertainty $\sigma^2_{\text{ep}}(x, z)$.
3. **Aleatoric Uncertainty**: Sampling from $p(y \,|\, x)$, the aleatoric uncertainty $\sigma^2_{\text{al}}(x)$ is estimated as the variance of the samples.

HybridFlow is trained in a two-stage process that reflects its hybrid architecture. First, the normalizing flow (NF) component is trained independently to learn the conditional distribution $p(y \,|\, x)$, which is used to quantify aleatoric uncertainty. The NF is optimized using the NLL loss of the conditional distribution (Equation 3), a standard and widely adopted objective for training normalizing flows to model the full data distribution. After training the flow, the latent representation $z = f^{-1}(y \,|\, x)$ is extracted and used as an additional input to the predictor model, along with the original input $x$. Crucially, because the inverse mapping of the flow is conditioned on $x$, the resulting $z$ encodes structured information about the conditional distribution, rather than being independent noise. This process ensures $z$ is a meaningful representation, distinguishing it from the generative use of the flow (Step 3), where $y$ is sampled from the learned posterior distribution $p(y|x)$ to estimate aleatoric uncertainty (see Appendix B). During the second stage, the predictor model is trained using a task-appropriate loss function, such as mean squared error (MSE), and optionally paired with a Bayesian approximation method to estimate epistemic uncertainty.

The NF can be optionally fine-tuned during the training of the predictor model to allow improved joint representations, though in most applications the flow converges independently and does not require additional updates. Similarly, the predictor model can be fine-tuned after integration with the NF outputs. HybridFlow

supports any probabilistic predictor architecture for epistemic uncertainty estimation; in our experiments, we use MC Dropout for simplicity and compatibility with prior work, but the framework can accommodate deep ensembles, variational inference, or other Bayesian or Bayesian-approximation techniques.

A key consideration when implementing HybridFlow is the dimensionality of the target variable $y$, which constrains the use of normalizing flows. High-dimensional output spaces increase computational and memory requirements due to the need for tractable Jacobian determinants in the NF. In such cases, the NF can be paired with a feature extractor or autoencoder to first compress the output space into a lower-dimensional latent representation, as we demonstrate in Section 4.1. This preserves the flow's modeling capacity while ensuring that the method remains tractable for large-scale or high-resolution outputs. In practice, we find that moderate-sized flows (4–6 layers) suffice for the regression tasks we tested, with deeper flows offering diminishing returns (see Appendix C for ablation study). For more complex datasets, additional flow depth or hidden capacity may improve performance, but at a computational cost that scales approximately linearly with the number of flow layers and size of the flow output (Papamakarios et al., 2021; Bhattacharyya et al., 2020).

## 4 Experiments

This section presents the evaluation of the HybridFlow architecture on various distinct datasets, including the NYU Depth v2 dataset for depth estimation and 12 UCI regression benchmarks. The experiments focus on assessing both predictive accuracy and uncertainty quantification capabilities. We benchmark HybridFlow against a baseline model based on the framework of Kendall & Gal (2017), a similar framework from Seitzer et al. (2022) using the Beta-NLL (BNLL) loss, as well as a CBDL model, a state-of-the-art model-specific uncertainty quantification framework (Caprio et al., 2024).

### 4.1 Depth Estimation

We use the NYU Depth v2 dataset (Silberman et al., 2012), which is widely used for evaluating models in computer vision tasks, and the primary benchmark in Kendall & Gal (2017). It consists of 400,000 high-dimensional RGB and depth image pairs, covering a variety of environments such as offices, homes, and classrooms. For our experiments, we used a preprocessed subset comprising 795 training images and 654 test images, following the standard splits used in previous studies. Each RGB image is paired with a depth map representing the ground truth distances of each pixel from the camera. The diversity and complexity of the scenes, combined with the inherent measurement noise in depth sensors, make this dataset ideal for evaluating both aleatoric and epistemic uncertainty.

We train four models for quantifying aleatoric and epistemic uncertainty, two baseline heteroscedastic regression models, one with a Gaussian NLL loss (Kendall & Gal, 2017), another with a Beta-NLL loss (Seitzer et al., 2022), a deep ensemble of heteroscedastic Gaussian NLL loss models Lakshminarayanan et al. (2017), and a hybrid regression model with the proposed HybridFlow architecture. For the heteroscedastic regression models, we include a 2D Convolutional layer at the end of the predictor model to output the mean (prediction) and variance (aleatoric uncertainty) and train with their respective NLL-based losses. The deep ensemble of heteroscedastic regression models includes an ensemble of 5 Gaussian NLL models with the same architecture as the single-model NLL baseline. Each model was initialized and tuned following practices outlined in Lakshminarayanan et al. (2017). For the hybrid regression model, we use a CMAF paired with a UNet-based autoencoder to quantify the aleatoric uncertainty, sampling $p(y\,|\,x)$ with a sample size of 100 and computing the variance (following Section 3.3). While the autoencoder is not a necessary component of HybridFlow, we use an autoencoder to compress the input images (which are $384 \times 512 \times 3$) into a latent space of size 256, thus demonstrating the proposed approach is computationally manageable when working with high-dimensional data (for more information on the autoencoder, see Appendix A). We then use the generated latent representation from the CMAF and the RGB image as inputs to the predictor model.

For all experiments, we use a predictor model presented in Ganj et al. (2025), which achieves state-of-the-art performance on the NYU Depth v2 dataset across key metrics such as Absolute Relative Error (AbsRel) and Root Mean Squared Error (RMSE). Following Kendall & Gal (2017), we implement MC Dropout (Gal &

Ghahramani, 2016) to make the Ganj et al. (2025) predictor probabilistic for the quantification of epistemic uncertainty. For both Depth Regression and the UCI benchmarks we choose MC Dropout due to the simplicity of incorporating into existing models and to be consistent with the implementations of the baseline model in Kendall & Gal (2017). However, we include a comparison to a deep ensemble (Lakshminarayanan et al., 2017) of heteroscedastic regression models, to showcase that the performance improvements from the hybrid architecture will still be seen, regardless of the specific approach for quantifying epistemic uncertainty. For reference, we have also included a non-probabilistic version of the Ganj et al. (2025) model (without the ability to quantify uncertainty) in Table 1 in order to compare the effect that quantifying uncertainty has on the predictive performance. We also highlight that our implementation results show improvements over Kendall & Gal (2017), solely because we use a current state-of-the-art predictor model.

Despite its strong performance on a variety of uncertainty quantification tasks (Caprio et al., 2024), CBDL cannot be applied in this depth estimation task because the underlying UNet-based predictor for the depth estimation task is not a Bayesian model. As a result, CBDL's ensemble of variational guides cannot be constructed for a purely deterministic architecture, which limits its flexibility. This incompatibility represents a significant drawback of model-specific uncertainty frameworks when extending to non-Bayesian deep learning modules.

Training was conducted for 100 epochs using a batch size of 16, with the Adam optimizer (Kingma & Ba, 2015), a learning rate of 1e-4, and a weight decay of 1e-3. For the NF components, we used a CMAF with 5 flow layers and a conditional diagonal Gaussian base distribution using `nflows` (Durkan et al., 2020). We train the CMAF flow with a NLL loss prior to training the prediction model. Then, during the training stage of the predictor, we fine-tune the flow with a lower learning rate (1e-6) to integrate the latent representation $z$ into the hybrid architecture.

The probabilistic predictor for HybridFlow was configured with MC Dropout at a rate of 0.2 for epistemic uncertainty quantification, and we performed 50 stochastic forward passes at test time to approximate the predictive mean and epistemic variance. The heteroscedastic regression models minimized a Gaussian NLL and Beta-NLL losses, while the HybridFlow model minimized a combined loss consisting of the flow's negative log-likelihood, and a scale-invariant log loss (Eigen et al., 2014) and multi-scale gradient loss (Ganj et al., 2025) for the predictor.

All training was conducted on a single Tesla V100 GPU, with the heteroscedastic models requiring approximately 25 hours to converge and HybridFlow requiring 36 hours. The models were evaluated using the standard metrics for depth estimation, including MSE, AbsRel, and thresholds ($\delta_{1-3}$), which indicate the proportion of predictions within 25%, 56%, and 95% of the ground truth values (Eigen et al., 2014).

In this study, we include a robust analysis of the quantified uncertainty from the approaches tested. Following Wang et al. (2025), we evaluate calibration quality using the Expected Calibration Error (ECE), which measures the average discrepancy between the confidence intervals and predicted accuracy, and the Prediction Interval Coverage Probability (PICP) which reports the fraction of true values contained within the computed uncertainty intervals. To assess the sharpness of our uncertainty estimates, we compute the Mean Prediction Interval Width (MPIW), capturing the typical size of the predictive intervals, and the Winkler score, which penalizes both overly wide and under-covering intervals. Additionally, we apply strictly proper scoring rules (Gneiting & Raftery, 2007), such as the Negative Log Likelihood score (NLL) and the Continuous Ranked Probability Score (CRPS), which penalize miscalibration in the full predictive distribution. Total uncertainty, within the scope of this work, is the sum of the aleatoric and epistemic uncertainty, which is common in uncertainty quantification literature (Depeweg et al., 2018; Hüllermeier & Waegeman, 2021; Wimmer et al., 2023). Together, these metrics furnish a concise yet comprehensive characterization of aleatoric and epistemic uncertainty across predictive accuracy, calibration, sharpness, and reliability. Definitions for each uncertainty metric can be found in Appendix D.

## 4.2   UCI Regression Benchmarks

To evaluate the robustness and accuracy of HybridFlow, we compared it against the same heteroscedastic regression methods on the UCI regression datasets, as detailed in Section 4.1, and a CBDL model (Caprio et al., 2024). The UCI datasets are a widely recognized benchmark suite for regression tasks, offering diverse

data distributions, varying noise levels, and differing scales that are ideal for evaluating models' predictive accuracy and uncertainty estimation capabilities.

We follow a standard experimental protocol for Bayesian deep learning frameworks (Seitzer et al., 2022; Hernández-Lobato & Adams, 2015; Gal & Ghahramani, 2016), using the same datasets and splits. Each dataset was evaluated 20 times with random 80/20 splits for training and testing, normalizing input features and target outputs to zero mean and unit variance based on the training split. For more experimental details, see Seitzer et al. (2022).

We train a flow using the HybridFlow framework with a CMAF with 5 flow layers. We first train the flow alone using a NLL loss. Then, the latent representations generated by the flow were combined with the original inputs and fed into the predictor model, which incorporates MC Dropout for modeling epistemic uncertainty. The predictor model used for all three methods is a simple MLP with one hidden layer of 50 ReLU-activated units that is trained using an MSE loss and early stopping, as in Seitzer et al. (2022).

HybridFlow is compared to heteroscedastic regression models trained with a Gaussian NLL loss (Kendall & Gal, 2017) and Beta-NLL loss (Seitzer et al., 2022), as well as a non-probabilistic version of the model trained with an MSE loss. We conduct the experiments using a Tesla V100 GPU with an Adam optimizer and perform a grid search of possible learning rates ranging from 1e-3 to 1e-5. To quantify epistemic uncertainty, we applied MC Dropout with a dropout probability of 0.2 and performed 30 stochastic forward passes during evaluation. In HybridFlow, we sample the learned posterior distribution 100 times to estimate the aleatoric uncertainty learned by the CMAF.

In addition to the heteroscedastic regression baselines, we also benchmark against CBDL (Caprio et al., 2024). We implement CBDL as a credal set of Bayesian models in Pyro (Bingham et al., 2019) that closely resemble the MLP models used for HybridFlow and heteroscedastic regression baselines. We initialize four BNNs, each with one hidden layer of 50 Tanh activated units, by varying the Gaussian prior scale and likelihood over the parameters and train each of them using stochastic variational inference (SVI). Aleatoric and epistemic uncertainties are computed using individual model entropies and variability within the credal set as outlined in Caprio et al. (2024).

We evaluate the predictive accuracy of each model using the mean RMSE value and the standard deviation of the 20 variations of the model based on the random splits. We then evaluate the uncertainty quantification performance of each model using a robust selection of metrics that collectively capture critical aspects of uncertainty estimation including the NLL, ECE, CRPS, PICP, and MPIW, as described in Section 4.1.

## 5 Results

### 5.1 Depth Estimation Results

HybridFlow demonstrates superior predictive accuracy on the NYU Depth v2 dataset, outperforming baseline models that use heteroscedastic regression methods. As seen in Table 1, the HybridFlow model achieves improved results across several evaluation metrics, including MSE, AbsRel, and $\delta$ thresholds ($\delta_1$-$\delta_3$). The

Table 1: Comparison of hybrid depth estimation model with the heteroscedastic depth estimation models (with MC Dropout to quantify uncertainty) and a deep ensemble of NLL models using NYU Depth v2. The Ganj et al. (2025) model does not quantify uncertainty, but is included to compare the predictive accuracy changes when adding uncertainty quantification capabilities.

| Training Method | MSE | AbsRel | $\delta_1$ | $\delta_2$ | $\delta_3$ |
|---|---|---|---|---|---|
| NLL (Kendall & Gal, 2017) | 0.129 | 0.083 | 0.947 | 0.987 | 0.995 |
| BNLL (Seitzer et al., 2022) | 0.134 | 0.087 | 0.947 | 0.987 | 0.994 |
| Deep Ensemble | 0.166 | 0.100 | 0.927 | 0.985 | 0.994 |
| HybridFlow | **0.058** | **0.041** | **0.989** | **0.998** | **0.999** |
| Ganj et al. (2025) (no uq) | 0.057 | 0.039 | 0.989 | 0.998 | 0.999 |

Table 2: Uncertainty quantification metrics on the NYU Depth v2 datasets. Metrics include NLL (Negative Log-Likelihood), calibration error (ECE), sharpness, total PICP (Prediction Interval Coverage Probability), CRPS (Continuous Ranked Probability Score), and MPIW (Mean Prediction Interval Width).

| METHOD | NLL | | | ECE | | | WINKLER | | | MPIW | | | PICP | CRPS |
|---|---|---|---|---|---|---|---|---|---|---|---|---|---|---|
| | ALE. | EPI. | TOT. | ALE. | EPI. | TOT. | ALE. | EPI. | TOT. | ALE. | EPI. | TOT. | | |
| NLL | 5.79 | 10.88 | 7.39 | 4.42 | **2.98** | 3.07 | 14.53 | 11.51 | 10.16 | 5.38 | 0.035 | 4.04 | 0.97 | 2.71 |
| BNLL | 2.95 | 3.37 | 2.73 | 2.32 | 10.96 | 4.63 | 6.18 | 10.52 | 8.23 | **3.70** | 0.78 | **3.60** | 0.92 | 1.66 |
| DEEP ENSEMBLE | 6.10 | 10.22 | 8.18 | 3.70 | 12.25 | 4.38 | 10.02 | 9.78 | 9.35 | 5.93 | 0.061 | 3.87 | 0.91 | 3.17 |
| HYBRIDFLOW | **1.96** | **0.09** | **0.73** | **0.68** | 4.14 | **0.46** | **4.82** | 7.46 | **4.83** | 4.70 | **0.64** | 4.76 | **0.99** | **0.37** |

Hybrid model achieves an MSE of 0.058 and an AbsRel of 0.041, which are improvements over the Gaussian NLL framework (MSE: 0.129, AbsRel: 0.083) and the Beta-NLL approach (MSE: 0.134, AbsRel: 0.087). Notably, the performance is on par with the Ganj et al. (2025) predictor model when uncertainty quantification is excluded (MSE: 0.057, AbsRel: 0.039), illustrating that the integration of uncertainty estimation does not compromise predictive accuracy. We also highlight that the choice of epistemic uncertainty quantification method does not greatly impact the predictive performance, as can be seen by the comparison of NLL results to the Deep Ensemble. We attribute the slightly higher error observed in the Deep Ensemble to known training instabilities of heteroscedastic NLL models (Seitzer et al., 2022; Stirn et al., 2023; Wong-Toi et al., 2024), where some ensemble members converge to suboptimal minima, and following the Deep Ensemble protocol (Lakshminarayanan et al., 2017), we retain these members rather than restarting or re-tuning them, which increases error relative to the best single model.

HybridFlow also exhibits robust uncertainty quantification capabilities, addressing both aleatoric and epistemic uncertainties effectively. The aleatoric uncertainty, modeled using a CMAF in the hybrid architecture, captures inherent noise in the dataset, while epistemic uncertainty is quantified through MC Dropout in the predictor model. As shown in Table 2, HybridFlow achieves consistently better aleatoric ECE and PICP with epistemic ECE on par with the other models, indicating that its uncertainty estimates are better calibrated than those of existing methods. It also shows improved performance across other uncertainty metrics, including lower CRPS and NLL, and more reliable predictive intervals as measured by the Winkler score. While HybridFlow has slightly wider uncertainty intervals on average (as reflected by MPIW), this trade-off supports better coverage and reliability.

Qualitative analyses of depth maps further support the quantitative findings, the HybridFlow model predicts sharper depth edges and maintains consistency across scenes with varied lighting and texture conditions (Appendix E). HybridFlow's uncertainty maps reveal elevated aleatoric uncertainty for areas with abnormal lighting or occlusion boundaries, consistent with regions expected to pose challenges due to aleatoric noise. For epistemic uncertainty, higher uncertainty is more localized to objects or textures that are underrepresented in the training dataset. We also see that the aleatoric uncertainty predicted by the heteroscedastic methods closely resembles the predicted depth, which may indicate that the model is unable to effectively learn noise distributions at variable depths.

## 5.2 UCI Regression Benchmarks Results

Table 3 shows that the HybridFlow models consistently are more accurate across the suite of UCI regression benchmarks. Based on RMSE, out of the 12 UCI regression benchmarks we tested, the HybridFlow models outperform the Gaussian NLL, BNLL, and CBDL models on 9 of the datasets. This highlights HybridFlow's ability to excel in diverse conditions, outperforming the other methods in both high-dimensional settings and tasks characterized by challenging noise distributions.

The NF not only allows the model to quantify aleatoric uncertainty, but in 6 of the benchmarks the HybridFlow model achieves even greater accuracy than the non-probabilistic models. Of the remaining 6 benchmarks, 5 show RMSE values within 5% of the non-probabilistic MLP baseline. This suggests that the latent feature representations learned by the NF contribute meaningful additional structure to the data,

Table 3: RMSE values for each of the UCI regression datasets from a standard MLP model compared to heteroscedastic training methods (NLL, BNLL), a deep ensemble baseline, and a non-probabilistic predictor. $N$, $d$, and $k$ represent the dataset length, input dimensions, and output dimensions respectively.

| Dataset | $N$ | $d$ | $k$ | NLL | BNLL | CBDL | Deep Ensemble | HybridFlow | MLP (no uq) |
|---|---|---|---|---|---|---|---|---|---|
| Boston Housing | 506 | 13 | 1 | $3.56 \pm 1.07$ | $3.42 \pm 1.04$ | $3.40 \pm 0.74$ | $3.82 \pm 1.12$ | $\mathbf{3.12 \pm 0.85}$ | $3.24 \pm 1.08$ |
| Carbon | 10721 | 5 | 3 | $0.0068 \pm 0.003$ | $0.0068 \pm 0.003$ | $0.0076 \pm 0.002$ | $0.0072 \pm 0.003$ | $\mathbf{0.0068 \pm 0.002}$ | $0.0068 \pm 0.003$ |
| Concrete Strength | 1030 | 8 | 1 | $6.08 \pm 0.65$ | $5.61 \pm 0.65$ | $5.52 \pm 0.22$ | $6.32 \pm 0.71$ | $\mathbf{5.07 \pm 0.52}$ | $4.96 \pm 0.64$ |
| Cycle Power Plant | 9568 | 4 | 1 | $4.06 \pm 0.18$ | $4.04 \pm 0.15$ | $4.26 \pm 0.20$ | $4.18 \pm 0.19$ | $\mathbf{4.02 \pm 0.18}$ | $4.01 \pm 0.19$ |
| Energy Efficiency | 768 | 8 | 2 | $2.25 \pm 0.34$ | $1.12 \pm 0.25$ | $\mathbf{0.97 \pm 0.08}$ | $1.30 \pm 0.22$ | $1.00 \pm 0.14$ | $0.92 \pm 0.11$ |
| Kin8m | 8192 | 8 | 1 | $0.087 \pm 0.004$ | $0.081 \pm 0.003$ | $0.84 \pm 0.002$ | $0.091 \pm 0.004$ | $\mathbf{0.078 \pm 0.003}$ | $0.081 \pm 0.003$ |
| Naval Propulsion | 11934 | 16 | 2 | $0.0021 \pm 0.0006$ | $\mathbf{0.0004 \pm 0.0001}$ | $0.026 \pm 0.0043$ | $0.0024 \pm 0.0007$ | $\mathbf{0.0004 \pm 0.0001}$ | $0.0004 \pm 0.0001$ |
| Protein Structure | 45730 | 8 | 1 | $4.49 \pm 0.11$ | $4.28 \pm 0.02$ | $4.91 \pm 0.11$ | $4.62 \pm 0.11$ | $\mathbf{4.26 \pm 0.05}$ | $4.28 \pm 0.07$ |
| Superconductivity | 21263 | 81 | 1 | $13.87 \pm 0.50$ | $13.02 \pm 0.47$ | $14.98 \pm 2.57$ | $14.42 \pm 0.56$ | $\mathbf{11.87 \pm 0.45}$ | $12.48 \pm 0.40$ |
| Wine Quality (Red) | 1599 | 11 | 1 | $0.636 \pm 0.038$ | $\mathbf{0.635 \pm 0.037}$ | $0.665 \pm 0.040$ | $0.652 \pm 0.041$ | $0.639 \pm 0.036$ | $0.633 \pm 0.036$ |
| Wine Quality (White) | 4898 | 11 | 1 | $0.691 \pm 0.032$ | $0.685 \pm 0.035$ | $0.737 \pm 0.037$ | $0.702 \pm 0.033$ | $\mathbf{0.678 \pm 0.030}$ | $0.684 \pm 0.038$ |
| Yacht Hydrodynamics | 308 | 6 | 1 | $\mathbf{1.22 \pm 0.47}$ | $1.73 \pm 1.00$ | $2.30 \pm 1.94$ | $1.41 \pm 0.61$ | $1.55 \pm 0.79$ | $0.78 \pm 0.25$ |

Table 4: Training run-time (in minutes wall-time) for each of the UCI regression datasets from a standard MLP model compared to heteroscedastic training methods (NLL, BNLL), a deep ensemble of heteroscedastic regression models, and a non-probabilistic predictor.

| Dataset | NLL | BNLL | CBDL | Deep Ensemble | HybridFlow | MLP (no uq) |
|---|---|---|---|---|---|---|
| Boston Housing | **0.19** | 0.21 | 10.91 | 0.95 | 1.11 | 0.37 |
| Carbon | 4.49 | 3.38 | 24.50 | 22.45 | 16.62 | **2.95** |
| Concrete Strength | **0.22** | 0.30 | 12.27 | 1.10 | 2.04 | 0.80 |
| Cycle Power Plant | 5.97 | 4.69 | 43.84 | 29.85 | 21.95 | **4.30** |
| Energy Efficiency | **0.75** | 0.89 | 17.15 | 3.75 | 2.63 | 1.19 |
| Kin8m | **2.94** | 3.28 | 29.51 | 14.70 | 18.43 | 2.99 |
| Naval Propulsion | 6.60 | 6.30 | 24.79 | 33.00 | 16.47 | **4.93** |
| Protein Structure | 6.05 | 6.10 | 44.83 | 30.25 | 28.59 | **4.75** |
| Superconductivity | 4.70 | **4.39** | 37.28 | 23.50 | 43.35 | 4.95 |
| Wine Quality (Red) | **0.30** | **0.30** | 7.12 | 1.50 | 2.73 | 0.32 |
| Wine Quality (White) | 0.83 | 0.82 | 24.03 | 4.15 | 8.61 | **0.66** |
| Yacht Hydrodynamics | **0.25** | 0.28 | 14.48 | 1.25 | 1.69 | 1.17 |

improving predictive performance beyond what is achievable with MSE optimization alone (for a detailed ablation study on HybridFlow inputs, see Appendix B, Table 7). While the model's performance surpasses others in most scenarios, a few cases where it matches the performance of baseline methods suggest the influence of dataset-specific complexities on results. Table 4 showcases the training run-times of each of the 4 models tested. Our results show HybridFlow does add to total training time, but less so than the equivalent Bayesian method, CBDL. However, we consider that a small trade-off when comparing with HybridFlow accuracy and argue that the extra training time provides enough practical benefit to justify its use.

Across the UCI suite HybridFlow showcases superior performance on nearly all uncertainty quantification metrics. It attains the lowest total NLL on 11 of 12 datasets, the lowest Winkler score on 11 of 12, and the smallest total ECE on 9 of 12 (Table 5; see Appendix F for full results). Despite these sharper and better-calibrated distributions, it still produces the narrowest intervals (lowest MPIW on 8 of 12 datasets) while preserving reliability (PICP remains near 0.97 on every benchmark). These results confirm that decoupling aleatoric and epistemic components with HybridFlow yields predictive uncertainties that are simultaneously sharp, calibrated, and trustworthy across nearly all of the UCI benchmarks. Additionally, we demonstrate that the uncertainties quantified by HybridFlow are disentangled, as can be seen in Appendix G.

# 6 Case Study: Hybrid Ice Sheet Emulator

To test the efficacy of HybridFlow in a real-world scenario and demonstrate how it could be implemented with an existing model structure, we use the HybridFlow framework to create an emulator for projecting ice sheet dynamics and their contributions to sea level rise. Ice sheet evolution is governed by complex, nonlinear processes, such as melting, snow accumulation, and ice shelf instability. These processes involve large amounts of uncertainty, which originate both from the physical modeling of the processes and from the chaotic nature of the processes themselves. Ice sheet emulators (Van Katwyk et al., 2023; Edwards et al.,

Table 5: Total uncertainty quantification metrics across models on the UCI datasets. Metrics include total NLL (Negative Log-Likelihood), total calibration error (ECE), total Winkler score, total MPIW (Mean Prediction Interval Width), PICP (Prediction Interval Coverage Probability), and CRPS (Continuous Ranked Probability Score). An expanded version of this table, which includes metrics for aleatoric, epistemic, and total uncertainty is included in Table 10 in Appendix F.

| DATASET | METHOD | NLL | ECE | WINKLER | MPIW | PICP | CRPS |
|---|---|---|---|---|---|---|---|
| BOSTON HOUSING | NLL | 2.88 | 2.40 | 86.35 | 86.32 | **1.00** | 1.98 |
| | BNLL | 1.98 | **1.88** | 33.43 | 32.94 | **1.00** | **1.65** |
| | DEEP ENSEMBLE | 2.20 | 2.05 | 48.50 | 45.00 | 0.99 | 1.80 |
| | CBDL | 5.97 | 200.15 | 769.99 | 662.91 | 0.94 | 1.83 |
| | HYBRIDFLOW | **1.63** | 1.90 | **19.65** | **16.63** | 0.98 | 1.76 |
| CARBON | NLL | **-2.92** | 21.06 | **0.20** | **0.19** | **1.00** | **0.01** |
| | BNLL | -2.79 | 17.81 | 0.23 | 0.23 | **1.00** | 0.02 |
| | DEEP ENSEMBLE | -2.85 | 18.90 | 0.22 | 0.22 | **1.00** | 0.02 |
| | CBDL | 345.93 | **4.39** | 36.61 | 0.98 | 0.89 | 0.03 |
| | HYBRIDFLOW | 52.15 | 23.90 | 0.26 | 0.21 | 0.99 | 0.02 |
| CONCRETE STRENGTH | NLL | 3.65 | 4.51 | 177.13 | 177.13 | **1.00** | 3.58 |
| | BNLL | 3.46 | **4.16** | 137.57 | 137.57 | **1.00** | 3.29 |
| | DEEP ENSEMBLE | 3.55 | 4.30 | 150.00 | 145.00 | **1.00** | 3.40 |
| | CBDL | 6.97 | 588.65 | 257.75 | 348.11 | 0.98 | **3.22** |
| | HYBRIDFLOW | **2.40** | 4.44 | **40.83** | **39.51** | 0.99 | 3.76 |
| CYCLE POWER | NLL | 3.10 | 3.23 | 89.61 | 89.62 | **1.00** | **2.64** |
| | BNLL | 3.07 | 3.22 | 86.03 | 86.03 | **1.00** | **2.64** |
| | DEEP ENSEMBLE | 3.12 | 3.18 | 88.00 | 87.00 | **1.00** | 2.65 |
| | CBDL | 55.74 | 770.32 | 251.01 | 436.43 | 0.72 | 2.91 |
| | HYBRIDFLOW | **2.02** | **3.10** | **24.61** | **22.97** | 0.99 | 2.66 |
| ENERGY EFFICIENCY | NLL | 2.90 | 2.26 | 83.42 | 83.42 | **1.00** | 1.86 |
| | BNLL | 1.89 | 1.61 | 36.60 | 36.60 | **1.00** | 1.46 |
| | DEEP ENSEMBLE | 2.20 | 1.80 | 40.00 | 38.00 | **1.00** | 1.42 |
| | CBDL | 6.69 | 189.01 | 156.14 | 521.30 | 0.65 | 1.48 |
| | HYBRIDFLOW | **1.43** | **1.18** | **19.28** | **19.28** | **1.00** | **1.16** |
| KIN8M | NLL | -1.25 | 14.50 | 0.84 | 0.34 | 0.79 | 0.08 |
| | BNLL | -1.37 | 15.47 | 0.69 | **0.32** | 0.81 | 0.07 |
| | DEEP ENSEMBLE | -1.33 | 13.00 | 0.75 | 0.33 | 0.85 | 0.07 |
| | CBDL | 363.54 | 146.19 | 10.64 | 0.33 | 0.84 | 0.07 |
| | HYBRIDFLOW | **-1.86** | **8.42** | **0.51** | 0.50 | **0.99** | **0.06** |
| NAVAL PROPULSION | NLL | -2.89 | 0.29 | 0.59 | 0.26 | 0.98 | 0.02 |
| | BNLL | -2.33 | 0.41 | 0.62 | 0.28 | 0.97 | 0.03 |
| | DEEP ENSEMBLE | -2.80 | 0.30 | 0.58 | 0.25 | 0.98 | 0.02 |
| | CBDL | 155.82 | 261.39 | 0.51 | 0.40 | 0.88 | **0.01** |
| | HYBRIDFLOW | **-3.32** | **0.19** | **0.48** | 0.22 | **0.99** | 0.02 |
| PROTEIN STRUCTURE | NLL | 3.37 | 4.40 | 115.13 | 115.13 | **1.00** | 4.11 |
| | BNLL | 3.02 | 3.66 | 88.05 | 87.96 | **1.00** | 3.32 |
| | DEEP ENSEMBLE | 3.20 | 3.80 | 95.00 | 90.00 | 0.99 | 3.40 |
| | CBDL | 4.90 | 48.17 | 487.74 | 487.74 | **1.00** | **3.00** |
| | HYBRIDFLOW | **1.95** | **3.30** | **22.74** | **20.04** | 0.97 | 3.20 |
| SUPERCONDUCTIVITY | NLL | 6.06 | 14.55 | 2033.21 | 2033.23 | **1.00** | 12.49 |
| | BNLL | 4.33 | 8.12 | 647.08 | 646.82 | **1.00** | 6.51 |
| | DEEP ENSEMBLE | 4.80 | 9.00 | 700.00 | 680.00 | 0.99 | 6.80 |
| | CBDL | 9.47 | 157.99 | 1770.72 | 5461.87 | **1.00** | 7.60 |
| | HYBRIDFLOW | **2.74** | **7.77** | **63.02** | **58.80** | 0.99 | **6.40** |
| WINE QUALITY (RED) | NLL | 2.09 | 1.38 | 3.41 | 2.19 | 0.91 | 0.46 |
| | BNLL | 0.25 | 1.95 | 3.70 | **1.86** | 0.86 | **0.43** |
| | DEEP ENSEMBLE | 0.30 | 1.50 | 3.50 | 2.00 | 0.90 | 0.45 |
| | CBDL | 47.26 | 2.65 | 137.57 | 2.22 | 0.76 | **0.43** |
| | HYBRIDFLOW | **0.05** | **0.95** | **3.27** | 2.85 | **0.97** | 0.44 |
| WINE QUALITY (WHITE) | NLL | 0.16 | 1.20 | 3.54 | 2.42 | 0.92 | 0.48 |
| | BNLL | 0.16 | 1.28 | 3.53 | **2.37** | 0.91 | **0.47** |
| | DEEP ENSEMBLE | 0.17 | 1.10 | 3.52 | 2.40 | 0.93 | 0.48 |
| | CBDL | 25.18 | 3.78 | 147.42 | 3.04 | 0.64 | 0.49 |
| | HYBRIDFLOW | **0.13** | **0.75** | **3.48** | 3.14 | **0.97** | 0.48 |
| YACHT HYDRODYNAMICS | NLL | 1.31 | 4.29 | 6.58 | 9.61 | **1.00** | 0.28 |
| | BNLL | 1.25 | 3.11 | 5.52 | 8.16 | **1.00** | 0.25 |
| | DEEP ENSEMBLE | 1.28 | 3.50 | 5.80 | 8.80 | 0.99 | 0.26 |
| | CBDL | 30.11 | 15.50 | 183.31 | 40.83 | 0.71 | 0.30 |
| | HYBRIDFLOW | **0.62** | **2.19** | **3.17** | **3.47** | 0.98 | **0.22** |

Table 6: Comparison of performance metrics for three ice sheet emulators: HybridFlow, a Gaussian Process, and a non-probabilistic predictor (Deep Ensemble of LSTMs without uncertainty quantification), evaluated on the Antarctic Ice Sheet (AIS) and Greenland Ice Sheet (GrIS).

|      | EMULATOR          | MSE  | ECE  | PICP |
|------|-------------------|------|------|------|
| AIS  | HYBRIDFLOW        | **1.20** | **0.02** | **0.95** |
|      | GAUSSIAN PROCESS  | 3.05 | 0.08 | 0.90 |
|      | LSTM (NO UQ)      | 1.09 | –    | –    |
| GRIS | HYBRIDFLOW        | **1.02** | **0.01** | **0.96** |
|      | GAUSSIAN PROCESS  | 9.87 | 0.12 | 0.91 |
|      | LSTM (NO UQ)      | 0.87 | –    | –    |

2021) provide efficient and reliable approximation of the complex physics-based models, which allows ice sheet scientists to perform sensitivity testing to understand the effect that climate variables have on sea level projections, and enables more experimentation to und

Recent work has demonstrated the effectiveness of LSTM-based ice sheet emulators (Van Katwyk et al., 2023) in improving predictive accuracy over Gaussian Process-based approaches (Edwards et al., 2021) on the ISMIP6 ice sheet model projection dataset (Nowicki et al., 2016). Building on these advancements, we use a CMAF and a Deep Ensemble (Lakshminarayanan et al., 2017) of LSTM models to effectively model the temporal structure of ice sheet projections from 2015 to 2100 (see Appendix H for comparison to MC Dropout). We compare the HybridFlow emulator to a Gaussian Process emulator based on the individual projection performance as well as the ability to approximate the full range of ensemble predictions. Consistent with the approach in Section 4, we train a non-probabilistic emulator using the LSTM model based on Van Katwyk et al. (2023).

Table 6 shows that the HybridFlow framework achieves superior performance in both test accuracy and the ability to reproduce the full range of ensemble projections compared to the Gaussian Process-based ice sheet emulator. HybridFlow performs similarly to the non-probabilistic predictor in test accuracy (MSE), highlighting the framework's ability to deliver reliable and actionable projections in the context of real data. Furthermore, the framework's ability to generate calibrated (PICP) and accurate (ECE) uncertainty estimates ensures that it can provide confidence intervals suitable for policy and planning decisions.

## 7 Conclusion

In this work we introduce HybridFlow, a novel hybrid architecture that combines the strength of NFs and probabilistic prediction models to quantify both aleatoric and epistemic uncertainty within a single unified model. By leveraging the flexibility of NFs to model data-specific aleatoric uncertainty and incorporating probabilistic models for epistemic uncertainty estimation, HybridFlow achieves robust and calibrated predictions without compromising predictive accuracy. HybridFlow's design addresses two gaps in uncertainty quantification methods: the decrease in model predictive accuracy to quantify uncertainty and the miscalibration of current state-of-the-art uncertainty quantification methods (Seitzer et al., 2022). HybridFlow achieves this by decoupling the loss functions used for aleatoric and epistemic uncertainty estimation, avoiding the pitfalls of joint NLL-based training methods. Decoupling the loss functions allows for task-specific loss functions for predictors, which results in predictive accuracy levels comparable to non-probabilistic models while maintaining the ability to quantify and separate sources of uncertainty. We demonstrate the ability to consistently achieve better metrics for accuracy (RMSE) and uncertainty quantification (ECE) on a variety of regression tasks, including depth estimation and a series of standard benchmarks. Furthermore, we present a case study that demonstrates the ability of the HybridFlow framework to be effective in real-world applications, offering accurate emulation of complex systems such as continental-scale ice sheets.

HybridFlow offers a particularly practical solution for domain scientists and researchers who seek to incorporate robust uncertainty quantification into existing modeling workflows without changing their entire system. In many applied settings, such as Earth system modeling or ice sheet forecasting (Section 6), pre-

dictive frameworks are already well established and finely tuned (Irrgang et al., 2021); replacing them with task-specific uncertainty models is often infeasible. HybridFlow avoids this disruption by providing a modular framework that can be integrated with existing probabilistic predictors, enabling calibrated and interpretable estimates of both aleatoric and epistemic uncertainty with minimal architectural changes. This makes HybridFlow especially valuable for scientific applications where improving the transparency and trustworthiness of predictions is critical, but where flexibility and compatibility with legacy models remain essential.

HybridFlow's modular design ensures adaptability to a wide range of tasks, allowing for future advancements in robust and trustworthy AI systems. However, a key drawback of this method is the added computation required for implementation. In the test cases presented in this study, the HybridFlow model for depth estimation took 44% longer to train than the NLL-based models. Similarly, Table 4 shows that HybridFlow was also slower to train for the UCI datasets, as the NF must be trained before the training of the predictor models. However, HybridFlow requires less computational overhead than the CBDL method (as can be seen in Table 4), which took one or two orders of magnitude longer than both the NLL methods and HybridFlow to train, but still remains a potential drawback of HybridFlow. The NLL and BNLL methods, on the other hand, are simple to implement and the added ability to quantify aleatoric uncertainty requires a negligible increase in computation to add to any model. Therefore, future work may focus both on expanding the framework's scalability and exploring alternative generative modeling techniques, including Bayesian NFs, to further enhance its versatility, precision, and reliability in high-dimensional or complex data environments.

While HybridFlow provides an approach to separately estimate aleatoric and epistemic uncertainty, it is important to acknowledge that disentangling these two sources of uncertainty is inherently difficult in practice. Despite conceptual distinctions, they are often entangled in complex models and real-world data, and there is currently no definitive or universally accepted solution to completely separate them (Smith et al., 2024; de Jong et al., 2024; Valdenegro-Toro & Mori, 2022). Attempts to decompose uncertainty are still valuable, as demonstrated in our ice sheet case study, where understanding the source of uncertainty informs scientific and policy decisions. Even imperfect separation can provide useful insight into model limitations and the nature of data variability.

Future work should also include the investigation of epistemic uncertainty within the NF itself, as it is not explicitly modeled in the current implementation. Although this practice aligns with conventional flow-based modeling approaches, incorporating methods such as flow ensembles (Berry & Meger, 2023) could enhance the robustness of aleatoric uncertainty estimates by capturing model-level epistemic uncertainty, even though due to the size of the NF model, the epistemic uncertainty is likely small. Future work may also include the rigorous testing of this framework on a variety of classification tasks, with a comparison to Sale et al. (2024), including a detailed analysis of quantified uncertainty.

## Broader Impact Statement

HybridFlow provides a practical and accessible framework for uncertainty quantification, enabling users to enhance the reliability of their models without significant architectural changes or specialized expertise. By prioritizing ease of integration, HybridFlow lowers the barrier to adopting robust uncertainty estimation, making it especially valuable for applications where trust and interpretability are critical.

Rather than focusing solely on novelty, this work emphasizes usability and flexibility—characteristics that support broader adoption across disciplines, including scientific modeling, environmental forecasting, and healthcare. While users should remain mindful of data limitations, HybridFlow equips them with a straightforward tool for improving model transparency and decision-making confidence, contributing to more trustworthy AI systems.

## Acknowledgments

This effort was funded by the Office of Naval Research (ONR), under Grant Number N00014-23-1-2729 as well as the National Science Foundation Graduate Research Fellowship Program under Grant 2040433. Computational resources and services required for this study were provided by the Center for Computation and Visualization at Brown University. We would also like to thank the three anonymous reviewers for their invaluable feedback on improving this work.

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

## Appendix

## A  Autoencoder for Depth Estimation

For the depth estimation experiments detailed in Section 4.1, we utilized an autoencoder to compress the dimensionality of RGB images from the NYU Depth v2 dataset. The autoencoder employs a UNet-inspired architecture with an encoder-decoder structure enhanced by skip connections. The encoder gradually reduces the spatial dimensions of the input images through six convolutional layers, each followed by batch normalization and ReLU activation. The bottleneck compresses the high-dimensional input ($384 \times 512 \times 3$) into a compact 256-dimensional latent representation. The decoder reconstructs the images using transposed convolutions, with skip connections from the encoder aiding in preserving spatial details. A sigmoid activation in the final layer ensures that pixel values are mapped to the range $[0, 1]$.

The autoencoder was trained using a composite loss function designed to ensure high-quality reconstruction and edge preservation. This loss combines Mean Squared Error (MSE) with the Structural Similarity Index Measure (SSIM) to focus on perceptual similarity, alongside an edge preservation term that penalizes differences in gradients between the input and reconstructed images. Training was conducted with the Adam optimizer at a learning rate of $1 \times 10^{-4}$, decayed by a factor of 0.5 based on validation loss, for a maximum of 50 epochs. Early stopping with a patience of 5 epochs was applied to prevent overfitting. Data augmentation, including random cropping, flipping, and brightness adjustments, was employed to improve generalization.

Extensive testing was conducted with latent dimensions of 512, 1024, 128, and 64, but it was observed that performance improvements plateaued beyond 256 dimensions, so we selected the 256-dimensional latent space to balance computational efficiency and reconstruction quality. On the NYU Depth v2 validation set, the autoencoder achieved an MSE of 0.012, SSIM of 0.943, and an edge loss of 0.005.

The latent representations generated by the autoencoder were used as inputs to the Conditional Masked Autoregressive Flow (CMAF) within the HybridFlow framework. This approach allowed for accurate modeling of aleatoric uncertainty while maintaining computational efficiency.

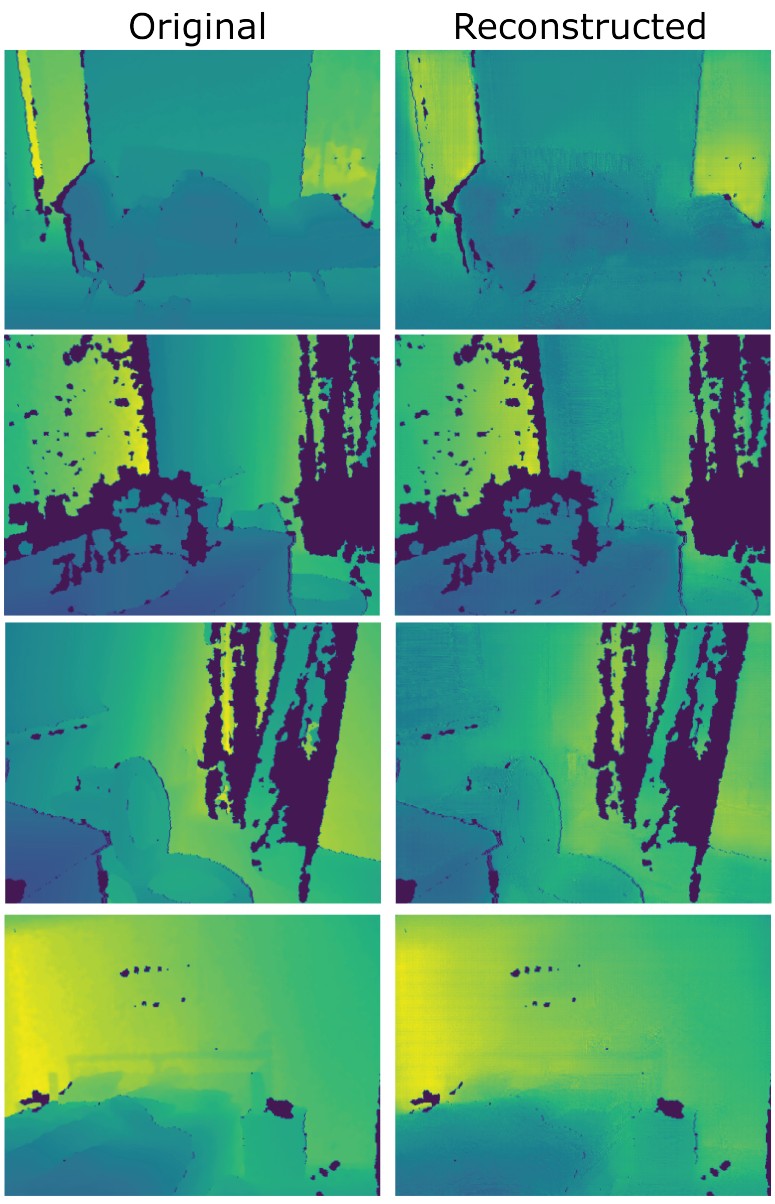

Figure 2: Reconstruction results from the autoencoder trained on the NYU Depth v2 dataset. Each column shows the true depth image (left) and the corresponding reconstructed depth map (right). The autoencoder effectively compresses high-dimensional input data into a 256-dimensional latent representation, enabling efficient dimensionality reduction while preserving fine-grained spatial details.

# B  Evaluating the Contribution of Latent Variable $z$ in HybridFlow

Table 7: Ablation study for inputs to the HybridFlow model, showing the RMSE values for each of the UCI regression datasets comparing the full HybridFlow model (predictor with both $x$ and $z$), an ablation with only $x$ or $z$ as input to the predictor, and a non-probabilistic MLP baseline.

| Dataset | $x$ and $z$ | $x$ only | $z$ only | Non-probabilistic, $x$ only |
|---|---|---|---|---|
| Boston Housing | **3.12 ± 0.85** | 3.26 ± 1.32 | 4.01 ± 1.12 | 3.24 ± 1.08 |
| Carbon | **0.0068 ± 0.0018** | 0.0068 ± 0.0024 | 0.0081 ± 0.0026 | 0.0068 ± 0.0028 |
| Concrete Strength | 5.07 ± 0.52 | **5.04 ± 0.77** | 5.61 ± 0.83 | 4.96 ± 0.64 |
| Cycle Power Plant | **4.02 ± 0.18** | 4.04 ± 0.18 | 4.59 ± 0.21 | 4.01 ± 0.19 |
| Energy Efficiency | **1.00 ± 0.14** | 1.15 ± 0.21 | 1.69 ± 0.23 | 0.92 ± 0.11 |
| Kin8m | **0.078 ± 0.003** | 0.082 ± 0.004 | 0.094 ± 0.005 | 0.081 ± 0.003 |
| Naval Propulsion | **0.0004 ± 0.0001** | 0.0011 ± 0.0004 | 0.0042 ± 0.0009 | 0.0004 ± 0.0001 |
| Protein Structure | **4.26 ± 0.05** | 4.33 ± 0.05 | 5.03 ± 0.07 | 4.28 ± 0.07 |
| Superconductivity | **11.87 ± 0.45** | 13.49 ± 0.53 | 14.88 ± 0.62 | 12.48 ± 0.40 |
| Wine Quality (Red) | 0.639 ± 0.036 | **0.637 ± 0.040** | 0.680 ± 0.045 | 0.633 ± 0.036 |
| Wine Quality (White) | **0.678 ± 0.030** | 0.701 ± 0.035 | 0.738 ± 0.038 | 0.684 ± 0.038 |
| Yacht Hydrodynamics | 1.55 ± 0.79 | **1.47 ± 0.58** | 2.38 ± 1.04 | 0.78 ± 0.25 |

To better understand the contribution of the latent representation $z$ from the normalizing flow to predictive performance, we conducted an ablation study across the UCI regression benchmarks. Table 7 compares four model inputs: the full HybridFlow model inputs (inputs $x$ and latent representation $z$), the HybridFlow architecture but only using the raw inputs $x$, the HybridFlow architecture but only using the computed $z$, and an MLP baseline (like that which was used for the HybridFlow predictor) but without uncertainty quantification capabilities. All other models use MC Dropout for uncertainty quantification, as explained in section 3.

This ablation helps isolate the impact of the NF-derived latent variable $z$ on performance, as well as disentangle improvements due to the hybrid architecture versus uncertainty quantification.

Table 7 shows that in most datasets, the model using the full HybridFlow inputs (both $x$ and $z$) achieves the lowest, or near-lowest RMSE. The combination appears especially beneficial for larger or more complex datasets, such as the Superconductivity, Protein Structure, and Naval Propulsion datasets, where the NF may extract structure not directly visible in the input space.

However, in a few datasets, the model trained only on raw inputs $x$ slightly outperforms the combined input model, indicating that the added latent structure may not always provide a benefit, particularly in smaller datasets. Models trained with $z$ alone consistently under-perform, confirming that the NF alone does not encode all information necessary for accurate prediction, further supportive the interpretation of $z$ as a complementary, uncertainty-aware feature rather than a standalone representation of the data.

## C   Computational Cost and Flow Depth Ablation

To better understand the computational tradeoffs associated with deeper flow-based models, we conducted an ablation study examining how the number of flow layers affects both training time and predictive performance. Table 8 reports the training time (in minutes) required for HybridFlow models using 2 to 12 flow layers across three representative UCI regression benchmarks: Housing, Concrete, and Energy. Table 9 reports the corresponding RMSE accuracy for each configuration.

As expected, training time increases with the depth of the flow model. This trend is particularly pronounced for higher-dimensional datasets like Energy and Concrete, where each additional layer adds nontrivial computational overhead. However, we observe diminishing returns (and even degradation) in predictive accuracy beyond a moderate number of flow layers. For all three datasets, RMSE is minimized between 4 and 6 flow layers. Deeper models (10+ layers) yield no consistent gains and can introduce training instability, especially for smaller datasets.

These results highlight a key practical consideration: while deeper normalizing flows can model more complex conditional distributions, they may not be necessary (or beneficial) for moderate-sized tabular datasets. For many real-world settings, a 4–6 layer flow offers a favorable balance between expressivity and efficiency.

Table 8: Training time (in minutes) for HybridFlow models with varying numbers of flow layers across three UCI datasets.

| Dataset | 2 layers | 4 layers | 6 layers | 8 layers | 10 layers | 12 layers |
|---|---|---|---|---|---|---|
| Housing | 1.01 | 1.49 | 1.92 | 2.69 | 2.71 | 3.11 |
| Concrete | 2.24 | 3.12 | 3.70 | 5.21 | 6.03 | 6.65 |
| Energy | 2.79 | 3.83 | 4.92 | 5.83 | 6.91 | 7.44 |

Table 9: RMSE accuracy for HybridFlow models with varying numbers of flow layers across three UCI datasets.

| Dataset | 2 layers | 4 layers | 6 layers | 8 layers | 10 layers | 12 layers |
|---|---|---|---|---|---|---|
| Housing | 3.11 | **2.56** | 2.71 | 2.93 | 2.87 | 3.34 |
| Concrete | 5.52 | **5.10** | 5.28 | 5.41 | 5.34 | 5.48 |
| Energy | 1.52 | **1.01** | 1.41 | 1.54 | 1.27 | 4.92 |

## D    Uncertainty Quantification Metrics

This appendix provides precise mathematical descriptions for the uncertainty quantification and evaluation metrics used in this paper.

### Expected Calibration Error (ECE)

The **Expected Calibration Error (ECE)** measures the difference between a model's predicted confidence and its empirical accuracy. For a model that outputs a predictive distribution, the prediction space is partitioned into $M$ bins, each corresponding to a range of probability values. The ECE is then computed as a weighted average of the absolute difference between the mean predicted probability and the fraction of positive outcomes in each bin:

$$\text{ECE} = \sum_{m=1}^{M} \frac{|B_m|}{N} \left| \text{acc}(B_m) - \text{conf}(B_m) \right| \tag{5}$$

where $N$ is the total number of samples, $B_m$ is the set of predictions in bin $m$, $\text{acc}(B_m)$ is the accuracy of the predictions in bin $m$, and $\text{conf}(B_m)$ is the average confidence of the predictions in bin $m$. A perfectly calibrated model has an ECE of 0, with lower values indicating better alignment between predicted confidence and empirical accuracy.

### Prediction Interval Metrics

For a given confidence level $(1 - \alpha)$, a prediction interval is defined by a lower bound $L(\mathbf{x})$ and an upper bound $U(\mathbf{x})$. The following metrics evaluate the quality of these intervals:

### Prediction Interval Coverage Probability (PICP)

**PICP** is the proportion of true target values $y_i$ that fall within their predicted interval $[L(\mathbf{x}_i), U(\mathbf{x}_i)]$. Ideally, for a $(1 - \alpha)$ prediction interval, the PICP should be close to $(1 - \alpha)$.

$$\text{PICP} = \frac{1}{N} \sum_{i=1}^{N} \mathbb{I}(y_i \in [L(x_i), U(x_i)]) \tag{6}$$

where $\mathbb{I}(\cdot)$ is the indicator function.

### Mean Prediction Interval Width (MPIW)

**MPIW** is the average width of the prediction intervals. Smaller MPIW values reflect narrower prediction intervals and correspond to sharper, more informative uncertainty estimates, assuming coverage is adequate.

$$\text{MPIW} = \frac{1}{N} \sum_{i=1}^{N} (U(x_i) - L(x_i)), \tag{7}$$

where $U(x_i)$ and $L(x_i)$ represent the upper and lower bounds of the predicted confidence interval for the i-th sample.

### Winkler Score

The **Winkler score** is a proper scoring rule that combines PICP and MPIW into a single metric. It penalizes wide intervals and intervals that do not contain the true value.

$$S_\alpha(L, U, y) = \begin{cases} (U - L) & \text{if } y \in [L, U] \\ (U - L) + \frac{2}{\alpha}(L - y) & \text{if } y < L \\ (U - L) + \frac{2}{\alpha}(y - U) & \text{if } y > U \end{cases} \tag{8}$$

The Winkler score rewards narrow intervals that contain the true value and penalizes both over-wide intervals and those that miss the target. Lower scores indicate better-calibrated and sharper predictive intervals.

**Probabilistic Forecasting Metrics**

**Negative Log-Likelihood (NLL)**

**NLL**, also known as the logarithmic score, evaluates the quality of a model's predicted probability distribution. It measures the likelihood of observed outcomes under a predicted distribution. For a model that outputs a probability density function $p(y|\mathbf{x})$, the NLL is:

$$\text{NLL} = -\frac{1}{N} \sum_{i=1}^{N} \log p(y_i|\mathbf{x}_i) \tag{9}$$

Lower NLL values indicate higher likelihood assigned to the observed outcomes under the predicted distribution, reflecting better-calibrated and more confident predictions.

**Continuous Ranked Probability Score (CRPS)**

The **CRPS** is a proper scoring rule that generalizes the Mean Absolute Error to probabilistic forecasts. It measures the difference between the predicted cumulative distribution function (CDF), $F$, and the empirical CDF of the observation.

$$\text{CRPS}(F, y) = \int_{-\infty}^{\infty} (F(z) - \mathbb{I}(y \leq z))^2 dz \tag{10}$$

where $F$ is the predicted CDF and $y$ is the observed value. Lower CRPS values correspond to predicted distributions that more closely match the observed outcome, in terms of both sharpness and calibration.

## E    Depth Estimation Visualizations

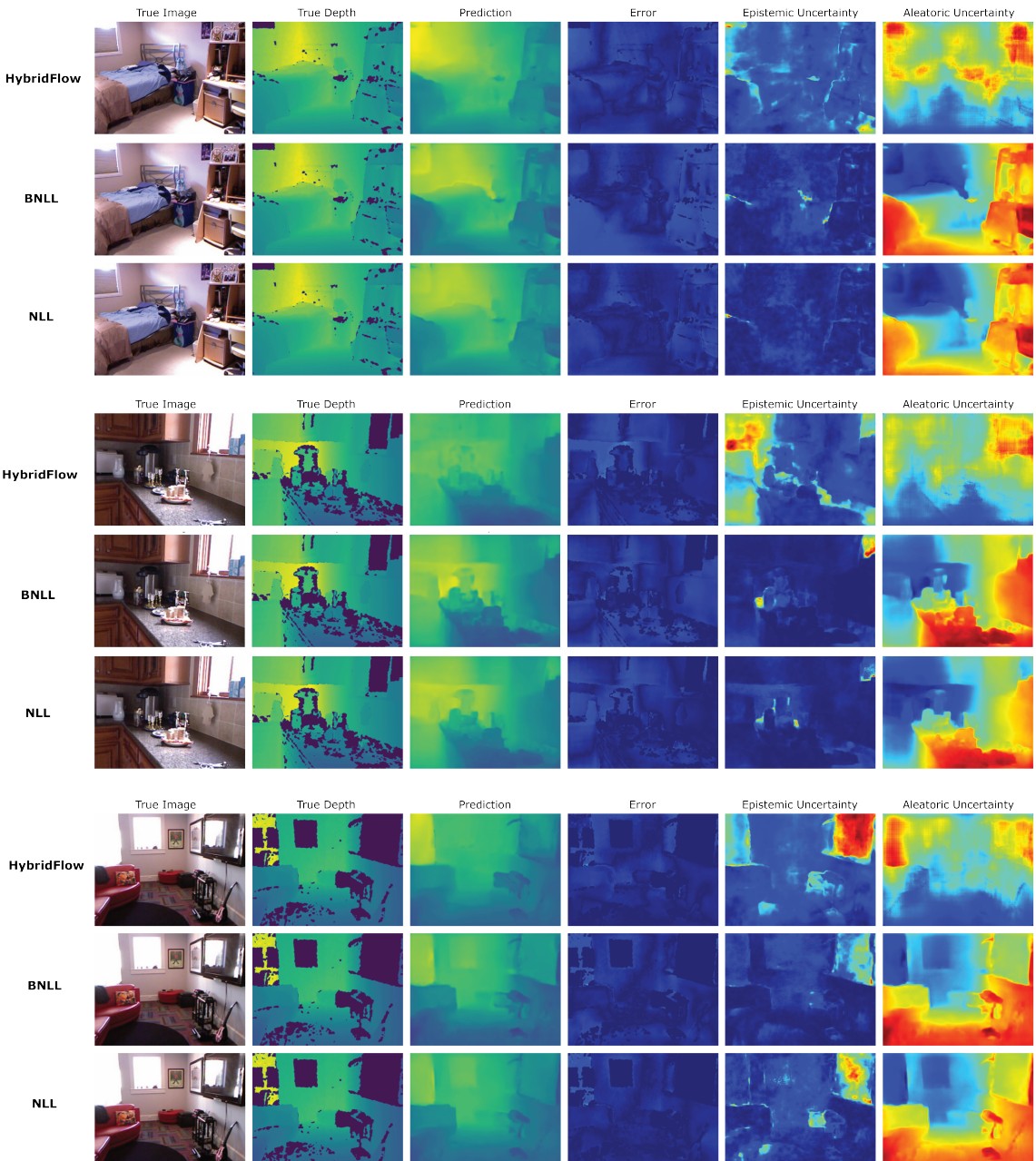

Figure 3: Visualization of depth estimation results using the HybridFlow framework compared to the BNLL (Seitzer et al., 2022) and NLL (Kendall & Gal, 2017) on the NYU Depth v2 dataset. Each row showcases a sample scene, with the columns representing (from left to right): the input RGB image, ground truth depth map, predicted depth map, model error, aleatoric uncertainty map, and epistemic uncertainty map. The predicted depth maps closely align with the ground truth, demonstrating accurate scene reconstruction. The aleatoric uncertainty maps highlight areas with inherent measurement noise, such as changes in image lighting (glare) or occlusion boundaries, while the epistemic uncertainty maps identify regions of the scene where the predictive model is uncertain. These visualizations illustrate the HybridFlow model's ability to provide both accurate predictions and reliable uncertainty quantification.

## F  Full UCI Uncertainty Results

Table 10: Uncertainty quantification metrics across models on the UCI datasets. Metrics include NLL (Negative Log-Likelihood), calibration error (ECE), Winkler score, MPIW (Mean Prediction Interval Width), total PICP (Prediction Interval Coverage Probability), and CRPS (Continuous Ranked Probability Score). Metrics are reported for each quantified aleatoric, epistemic, and total (additive) uncertainty.

| Dataset | Method | NLL Ale. | NLL Epi. | NLL Tot. | ECE Ale. | ECE Epi. | ECE Tot. | Winkler Ale. | Winkler Epi. | Winkler Tot. | MPIW Ale. | MPIW Epi. | MPIW Tot. | PICP | CRPS |
|---|---|---|---|---|---|---|---|---|---|---|---|---|---|---|---|
| Boston Housing | NLL | 2.80 | 4.11 | 2.88 | 2.39 | 1.79 | 2.40 | 80.82 | 35.17 | 86.35 | 80.59 | 5.73 | 86.32 | **1.00** | 1.98 |
| | BNLL | 2.39 | **3.15** | 1.98 | 1.83 | **1.38** | **1.88** | 29.96 | **28.28** | 33.43 | 27.94 | 5.00 | 32.94 | **1.00** | **1.65** |
| | CBDL | 491.23 | 6.28 | 5.97 | 188.00 | 199.59 | 200.15 | 5516.6 | 991.58 | 769.99 | 142.70 | 520.21 | 662.91 | 0.94 | 1.83 |
| | HybridFlow | **1.64** | 4.42 | **1.63** | **1.78** | 1.43 | 1.90 | **18.77** | 33.18 | **19.65** | **11.81** | **4.82** | **16.63** | 0.98 | 1.76 |
| Carbon | NLL | 43.71 | **-2.96** | **-2.92** | 39.28 | 22.44 | 21.06 | 0.61 | **0.19** | **0.20** | **0.01** | **0.18** | **0.19** | **1.00** | **0.01** |
| | BNLL | **23.53** | -2.86 | -2.79 | 27.32 | 19.40 | 17.81 | **0.46** | 0.21 | 0.23 | 0.02 | 0.21 | 0.23 | **1.00** | 0.02 |
| | CBDL | 370.94 | 317.27 | 345.93 | **19.62** | **4.29** | **4.39** | 7.92 | 37.46 | 36.61 | 0.02 | 0.96 | 0.98 | 0.89 | 0.03 |
| | HybridFlow | 24.63 | 7.68 | 52.15 | 35.24 | 22.21 | 23.90 | **0.46** | 0.24 | 0.26 | 0.02 | 0.19 | 0.21 | 0.99 | 0.02 |
| Concrete Strength | NLL | 3.56 | 4.03 | 3.65 | 4.51 | 4.12 | 4.51 | 165.80 | 60.02 | 177.13 | 165.80 | 11.33 | 177.13 | **1.00** | 3.58 |
| | BNLL | 3.36 | **3.80** | 3.46 | **4.15** | **3.76** | **4.16** | 126.25 | **54.34** | 137.57 | 126.25 | 11.32 | 137.57 | **1.00** | 3.29 |
| | CBDL | 148.05 | 7.07 | 6.97 | 588.89 | 587.23 | 588.65 | 1567.2 | 254.89 | 257.75 | 138.99 | 209.12 | 348.11 | 0.98 | **3.22** |
| | HybridFlow | **2.29** | 7.85 | **2.40** | 4.40 | 4.06 | 4.44 | **35.74** | 78.59 | **40.83** | **30.28** | **9.23** | **39.51** | 0.99 | 3.76 |
| Cycle Power | NLL | 3.03 | 5.45 | 3.10 | 3.22 | 2.56 | 3.23 | 82.94 | 49.66 | 89.61 | 82.94 | 6.68 | 89.62 | **1.00** | **2.64** |
| | BNLL | 3.00 | **5.30** | 3.07 | 3.22 | 2.57 | 3.22 | 79.40 | **49.41** | 86.03 | 79.39 | 6.64 | 86.03 | **1.00** | **2.64** |
| | CBDL | 135.34 | 60.87 | 55.74 | 577.36 | 770.71 | 770.32 | 306.80 | 253.72 | 251.01 | 142.80 | 293.63 | 436.43 | 0.72 | 2.91 |
| | HybridFlow | **1.93** | 6.53 | **2.02** | **3.03** | **2.51** | **3.10** | **20.53** | 51.85 | **24.61** | **16.62** | **6.35** | **22.97** | 0.99 | 2.66 |
| Energy Efficiency | NLL | 2.79 | 3.49 | 2.90 | 2.25 | 1.70 | 2.26 | 76.70 | 33.73 | 83.42 | 76.70 | 6.72 | 83.42 | **1.00** | 1.86 |
| | BNLL | 1.64 | 2.76 | 1.89 | 1.57 | 0.98 | 1.61 | 31.78 | 27.65 | 36.60 | 31.78 | **4.82** | 36.60 | **1.00** | 1.46 |
| | CBDL | 171.68 | 7.87 | 6.69 | 225.24 | 117.98 | 189.01 | 709.48 | 240.01 | 156.14 | 104.48 | 416.82 | 521.30 | 0.65 | 1.48 |
| | HybridFlow | **1.17** | **2.08** | **1.43** | **1.20** | **0.66** | **1.18** | **14.70** | **18.50** | **19.28** | **14.19** | 5.09 | **19.28** | **1.00** | **1.16** |
| Kin8m | NLL | 20.67 | 0.29 | -1.25 | 69.27 | **20.00** | 14.50 | 2.49 | 1.28 | 0.84 | 0.09 | 0.25 | 0.34 | 0.79 | 0.08 |
| | BNLL | 23.49 | **-0.60** | -1.37 | 76.36 | 20.01 | 15.47 | 2.32 | **0.91** | 0.69 | **0.07** | 0.25 | **0.32** | 0.81 | 0.07 |
| | CBDL | 2991.21 | 400.10 | 363.54 | 67.91 | 72.79 | 146.19 | 8.87 | 11.16 | 10.64 | 0.25 | **0.08** | 0.33 | 0.84 | 0.07 |
| | HybridFlow | **-1.94** | 1.76 | **-1.86** | **11.61** | 35.99 | **8.42** | **0.43** | 1.13 | **0.51** | 0.37 | 0.13 | 0.50 | **0.99** | **0.06** |
| Naval Propulsion | NLL | -3.24 | -0.60 | -2.89 | 0.26 | 0.18 | 0.29 | 0.39 | 1.03 | 0.59 | 0.05 | 0.21 | 0.26 | 0.98 | 0.02 |
| | BNLL | -2.61 | -0.54 | -2.33 | 0.38 | 0.27 | 0.41 | 0.46 | 0.95 | 0.62 | 0.06 | 0.22 | 0.28 | 0.97 | 0.03 |
| | CBDL | 201.38 | 132.27 | 155.82 | 778.69 | 182.37 | 261.39 | 162.60 | 473.69 | 487.74 | 0.05 | 0.35 | 0.40 | 0.88 | **0.01** |
| | HybridFlow | **-3.79** | **-0.47** | **-3.32** | **0.17** | **0.12** | **0.19** | **0.31** | **0.86** | **0.48** | **0.03** | **0.19** | **0.22** | **0.99** | 0.02 |
| Protein Structure | NLL | 3.34 | 63.83 | 3.37 | 4.40 | 2.61 | 4.40 | 112.45 | 133.39 | 115.13 | 112.45 | **2.68** | 115.13 | **1.00** | 4.11 |
| | BNLL | 2.97 | **6.01** | 3.02 | 3.66 | 2.28 | 3.66 | 84.57 | **96.59** | 88.05 | 84.45 | 3.51 | 87.96 | **1.00** | 3.32 |
| | CBDL | 92.66 | **4.88** | 4.90 | 40.91 | 47.15 | 48.17 | 162.60 | 473.69 | 487.74 | **14.05** | 473.69 | 487.74 | **1.00** | **3.00** |
| | HybridFlow | **1.97** | 38.88 | **1.95** | **3.26** | **1.84** | **3.30** | **21.54** | 97.91 | **22.74** | 17.17 | 2.87 | **20.04** | 0.97 | 3.20 |
| Superconductivity | NLL | 6.04 | 10.57 | 6.06 | 14.55 | 14.29 | 14.55 | 2013.1 | 133.96 | 2033.2 | 2013.14 | 20.09 | 2033.23 | **1.00** | 12.49 |
| | BNLL | **4.28** | **6.01** | 4.33 | 8.12 | 7.73 | 8.12 | 629.83 | **121.04** | 647.08 | 629.24 | 17.58 | 646.82 | **1.00** | 6.51 |
| | CBDL | 707.73 | 8.45 | 9.47 | 580.43 | 157.97 | 157.99 | 402.27 | 173.89 | 1770.7 | 3730.9 | 1730.8 | 5461.87 | **1.00** | 7.60 |
| | HybridFlow | 8.63 | 6.67 | **2.74** | **7.65** | **7.54** | **7.77** | **57.98** | 128.32 | **63.02** | **41.91** | **16.89** | **58.80** | 0.99 | **6.40** |
| Wine Quality (red) | NLL | 1.03 | 23.57 | 2.09 | 1.91 | 10.09 | 1.38 | 4.12 | 13.52 | 3.41 | 1.76 | **0.43** | 2.19 | 0.91 | 0.46 |
| | BNLL | 1.57 | **17.24** | 0.25 | 3.18 | 8.98 | 1.95 | 5.10 | **11.94** | 3.70 | **1.35** | 0.51 | **1.86** | 0.86 | **0.43** |
| | CBDL | 75.83 | 106.97 | 47.26 | 8.65 | **2.44** | 2.65 | 172.82 | 144.63 | 137.57 | 0.37 | 1.85 | 2.22 | 0.76 | **0.43** |
| | HybridFlow | **0.08** | 19.97 | **0.05** | **1.26** | 9.48 | **0.95** | **3.31** | 12.52 | **3.27** | 2.38 | 0.47 | 2.85 | **0.97** | 0.44 |
| Wine Quality (white) | NLL | 0.40 | 20.24 | 0.16 | 1.70 | 8.59 | 1.20 | 4.19 | 13.78 | 3.54 | 1.92 | 0.50 | 2.42 | 0.92 | 0.48 |
| | BNLL | 0.49 | **18.62** | 0.16 | 1.88 | 8.14 | 1.28 | 4.36 | **12.99** | 3.53 | **1.81** | 0.56 | **2.37** | 0.91 | **0.47** |
| | CBDL | 26.46 | 33.25 | 25.18 | 6.36 | **3.60** | 3.78 | 198.85 | 154.03 | 147.42 | 0.34 | 2.70 | 3.04 | 0.64 | 0.49 |
| | HybridFlow | **0.22** | 22.47 | **0.13** | **0.99** | 9.08 | **0.75** | **3.41** | 13.85 | **3.48** | 2.65 | **0.49** | 3.14 | **0.97** | 0.48 |
| Yacht Hydrodynamics | NLL | 1.21 | 4.33 | 1.31 | 4.07 | 3.53 | 4.29 | 6.03 | 5.57 | 6.58 | 6.30 | 3.31 | 9.61 | **1.00** | 0.28 |
| | BNLL | 1.35 | **3.98** | 1.25 | 3.10 | 2.54 | 3.11 | 5.36 | **4.03** | 5.52 | 5.66 | 2.50 | 8.16 | **1.00** | 0.25 |
| | CBDL | 29.11 | 32.17 | 30.11 | 10.99 | 18.83 | 15.50 | 138.35 | 168.73 | 183.31 | 25.39 | 15.44 | 40.83 | 0.71 | 0.30 |
| | HybridFlow | **0.63** | 5.00 | **0.62** | **2.01** | **1.51** | **2.19** | **2.59** | 5.80 | **3.17** | **2.26** | **1.21** | **3.47** | 0.98 | **0.22** |

# G    Disentanglement of Aleatoric and Epistemic Uncertainties

To further assess whether HybridFlow effectively separates aleatoric and epistemic uncertainty, we analyze their joint behavior across three regression datasets In Figure 4, we plot aleatoric uncertainty (x-axis) against epistemic uncertainty (y-axis) for all test samples across three methods: HybridFlow, standard Gaussian negative log-likelihood (NLL), and beta-NLL (BNLL). Each point represents a single test prediction.

HybridFlow exhibits dispersed epistemic values across narrow bands of aleatoric uncertainty, suggesting separation of the two uncertainty sources, while NLL displays more tightly coupled uncertainty values, with both types increasing together, especially in the Energy Efficiency dataset where a strong negative Spearman correlation is observed. BNLL shows partial disentanglement but still exhibits moderate correlation between the two uncertainties.

To quantify the level of dependency between aleatoric and epistemic uncertainty, we compute both Pearson's correlation coefficient ($r$) and Spearman's rank correlation coefficient ($\rho$) for each method and dataset. Spearman $\rho$ is emphasized due to its robustness to nonlinearity and non-Gaussian behavior.

Table 11: Correlation between Aleatoric and Epistemic Uncertainty (Spearman $\rho$ / Pearson $r$). Values closer to zero indicate greater disentanglement between aleatoric and epistemic uncertainty.

| Dataset | Method | Spearman $\rho$ | Pearson $r$ |
|---|---|---|---|
| Energy Efficiency | HybridFlow | **0.149** | **0.168** |
| | NLL | -0.334 | -0.324 |
| | BNLL | 0.240 | 0.182 |
| Boston Housing | HybridFlow | **0.333** | **0.373** |
| | NLL | 0.456 | 0.373 |
| | BNLL | 0.562 | 0.576 |
| Concrete Strength | HybridFlow | **0.147** | **0.235** |
| | NLL | 0.265 | 0.440 |
| | BNLL | 0.163 | 0.317 |

HybridFlow consistently demonstrates the lowest or near-lowest correlation between aleatoric and epistemic uncertainty across datasets. In contrast, NLL and BNLL exhibit stronger coupling, suggesting entanglement in their uncertainty estimates.

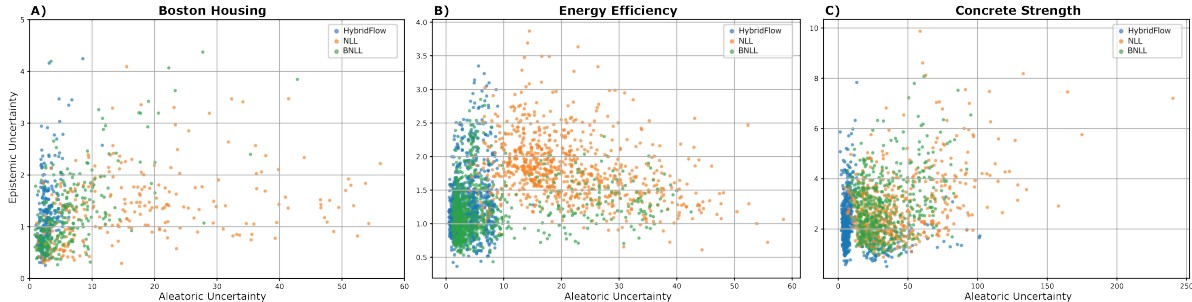

Figure 4: Scatter plots of aleatoric (x-axis) vs. epistemic (y-axis) uncertainty across test points for three datasets and three methods.

The smaller aleatoric intervals for HybridFlow in Boston Housing and Energy Efficiency are supported quantitatively by lower aleatoric MPIW/Winkler and aleatoric NLL (Table 10), while total PICP remains high. This aligns with the decoupled design: the flow models $p(y|x)$ and avoids absorbing model misspecification, which NLL/BNLL often express as inflated aleatoric variance (Seitzer et al., 2022).

# H Impact of Epistemic Uncertainty Method in HybridFlow Ice Sheet Emulator

HybridFlow is designed to be agnostic to the choice of epistemic uncertainty quantification method. In the main text, we adopt Monte Carlo Dropout (MCD) Gal & Ghahramani (2016) for its computational efficiency and ease of integration with the baseline architectures used in prior work. To verify that our conclusions for the ice sheet emulator case study are not specific to MCD, we conducted an ablation in the Ice Sheet Emulator case study (Sec. 6) where we compare MCD with the five-member Deep Ensemble (DE) Lakshminarayanan et al. (2017) as the epistemic uncertainty quantification method in HybridFlow.

Table 12 reports mean squared error (MSE), expected calibration error (ECE), and prediction interval coverage probability (PICP) for the Antarctic Ice Sheet (AIS) and Greenland Ice Sheet (GrIS) emulators. As expected, DE achieves slightly lower MSE and improved calibration compared to MCD, consistent with prior findings that ensembles produce stronger epistemic estimates. Crucially, the qualitative conclusions of the case study remain unchanged: HybridFlow yields well-calibrated predictions and high coverage under both epistemic methods, demonstrating that its benefit arises from the hybrid architecture rather than a specific epistemic estimator.

Table 12: Performance comparison between Monte Carlo Dropout (MCD) and Deep Ensembles (DE) in the Ice Sheet Emulator case study. Best values in bold.

|      | EMULATOR           | MSE      | ECE      | PICP     |
|------|--------------------|----------|----------|----------|
| AIS  | HYBRIDFLOW (DE)    | **1.20** | **0.02** | **0.95** |
|      | HYBRIDFLOW (MCD)   | 1.27     | 0.04     | 0.92     |
| GRIS | HYBRIDFLOW (DE)    | **1.02** | **0.01** | **0.96** |
|      | HYBRIDFLOW (MCD)   | 1.31     | 0.03     | 0.93     |

