# OpenReview forum: "HybridFlow: Quantification of Aleatoric and Epistemic Uncertainty with a Single Hybrid Model"
_TMLR — Accepted by TMLR_

### Review · Reviewer_ijDQ · 2025-07-23

**Summary Of Contributions:**

This paper proposes HybridFlow, a method to quantify uncertainty in regression models. HybridFlow is a two-step approach. In the first step, the authors train a conditional normalizing flow to learn the distribution of regression target $y$ given features $x$, $p(y|x)$. The flow has two purposes: first, computing the element-wise variance of its learned distribution provides an estimate of aleatoric uncertainty. Second, the flow produces a representation $z$ of $x$. In the second step, this representation is taken, along with $x$, to train a probabilistic model $p(y|x, z)$. The mean of this model is used for predictions, and its variance as an estimate of epistemic uncertainty.

The authors provide experiments on a depth estimation task where a depth scalar is predicted for every pixel of an image, UCI datasets, and on a task involving an ice sheet emulator.

Overall, I found the paper to be poorly written and the method to lack conceptual soundness.

**Audience:**

No

**Broader Impact Concerns:**

I have no broader impact concerns.

**Claims And Evidence:**

No

**Requested Changes:**

Please address the weaknesses mentioned above. In particular, for weakness 2, in addition to an explanation of why you believe the two models' variances are quantifying fundamentally different quantities (or of any misunderstanding I might have had), I would like to see scatterplots plotting your estimates of aleatoric and epistemic uncertainties. I suspect these estimates will tend to exhibit a very large correlation.

**Strengths And Weaknesses:**

## Strengths

1. The empirical evaluation considers various tasks and metrics to evaluate performance.

## Weaknesses

1. The paper is poorly written. It is overly verbose, yet some key pieces of the method are not clearly explained. In particular, the flow trained in the first step is meant to provide a representation $z$ of $x$, as per Figure 1. However, the flow takes $y$ as an input, so I do not understand how this representation can be computed without access to the ground truth $y$ which the model is trying to predict in the first place.

2. I also found the method to be poorly motivated, and I do not believe it disentangles aleatoric and epistemic uncertainties in the way claimed by the authors. In the second step, a model $p(y|x, z)$ is trained, but $z$ is meant to be a deterministic transformation of $x$. Since $x$ is also given as an input, adding $z$ to the conditioning adds no additional information, and so if the models in the first and second steps are flexible enough, they should learn the exact same distribution. To me this is a fundamental issue with the paper, since the authors claim that the variance of one model measures aleatoric uncertainty, while the variance of the other measures epistemic uncertainty.

3. Although the experiments consider enough tasks and metrics, they do not consider enough baselines. Deep ensembles are cited, but not compared against. I would like to see a comparison against an ensemble of heteroscedastic regression models, where aleatoric uncertainty is estimated as the average (over ensemble members) predicted variance, and where epistemic uncertainty is estimated as the variance (over ensemble members) of the mean prediction.

4. Lastly, a list of more minor points:
- The related work section should discuss deep ensembles; although this method is cited in the paper, it should not be cited in passing.
- Some statements should make it clear that the proposed quantities are estimates/measurements of aleatoric and epistemic uncertainty. For example, "the aleatoric uncertainty $\sigma_{al}^2(x)$ is calculated ..." makes it sounds like you are computing the ground truth aleatoric uncertainty.
- Although plenty of metrics are used, a precise mathematical description of every metric should be included in the appendix. As it stands, it is unclear what some of the entries in the tables represent. For example, what is the difference between the reported aleatoric and epistemic ECEs?
- It is unclear why deep ensembles are marked as "No UQ" in the ice sheet emulator experiments.

---

> ### Author Response · Authors · 2025-08-08
> **Response to Reviewer ijDQ**
>
> **Clarity & HybridFlow Inputs**
> Thank you for this insightful response. We recognize that this point may have been unclear. To clarify, the NF is trained on $(x, y)$ pairs to model the conditional distribution $p(y|x)$, and during training it uses both inputs and outputs. During inference time, the NF is used to sample the learned distribution $p(y|x)$, without requiring access to the ground truth $y$, as is standard practice for conditional generative models. The latent variable $z$ is used to extract representations during training, and samples of $y$ generated from $p(y|x)$ can be mapped to corresponding values of $z$ to train the predictor. We have revised and clarified the language in Section 3.1 to make this distinction between training and inference phases.
>
> **HybridFlow Motivation & Disentangling Uncertainties**
> This is a thoughtful point. The latent variable $z$ is derived as $z=f^{-1}(y|x)$, meaning that it captures the structural properties of the conditional distribution $p(y|x)$, including the data density of the outputs, given the information contained in $x$. While the predictor receives both $x$ and $z$, the inclusion of $z$ enables it to condition not only on the input but also on a representation of the distribution captured by the flow. To demonstrate this, we have included an ablation study of the HybridFlow inputs (Appendix F), which shows how the networks perform with $x$ and $z$ both as inputs, versus $x$ alone and $z$ alone. As the table shows, $z$ provides meaningful information when coupled with $x$.
>
> Addressing the comment on both networks quantifying the same uncertainty, HybridFlow’s design allows the predictor to separately learn epistemic uncertainty while incorporating the structured variability from the flow. These two sources of uncertainty are thus modeled by distinct components, each with different learning objectives and mechanisms. We note that the predictor and flow were trained on different loss functions, serve different purposes, and access different information: the flow estimates $p(y|x)$ to model aleatoric uncertainty, while the predictor uses $(x,z)$ to estimate point predictions and epistemic uncertainty.
>
> We agree that this subtlety may not have been sufficiently emphasized, and we have updated Section 3.2 to clarify the role that $z$ plays and how it may help the downstream predictor. We also add an appendix section to showcase the disentangling of uncertainties between methods.
>
> **Deep Ensembles as Baselines**
> Thank you for this suggestion. We agree that deep ensembles are a strong and widely-used method for estimating epistemic uncertainty. However, we would like to clarify that the primary focus of this work is not to benchmark different approaches for modeling epistemic uncertainty, but rather to evaluate how capturing aleatoric structure through a learned latent representation improves overall predictive and uncertainty performance. In our implementation of HybridFlow, we selected MC Dropout as the epistemic estimator due to its simplicity and consistency with prior work (e.g., Kendall & Gal, 2017). That said, we emphasize that HybridFlow is fully modular by design, and deep ensembles, or any other probabilistic predictor, can be substituted in place of MC Dropout without altering the core methodology. We have made changes to the manuscript to better discuss this point in Sections 3.2 and 4.1. Additionally, we have included an Appendix section comparing MC Dropout vs a Deep Ensemble in the Ice Sheet Emulator case study.
>
> With regard to an ensemble of heteroscedastic regression models, while this is a valid approach for capturing both aleatoric and epistemic uncertainty, it still relies on jointly training models using a single heteroscedastic loss. As discussed in Section 2, such joint loss functions have known limitations, including miscalibration and entanglement of uncertainty sources, that HybridFlow explicitly seeks to address through architectural separation. Therefore, although ensembles of heteroscedastic models may improve calibration to some extent, they do not avoid the foundational issues associated with joint modeling. We believe this distinction highlights a core contribution of HybridFlow, and we have clarified this in the updated manuscript.
>
> **Disentangling Uncertainties and Correlations**
> Thank you for this suggestion. To address this concern, we have added Appendix H, which includes scatter plots of the aleatoric vs epistemic uncertainty across three representative datasets, and report the Spearman and Pearson correlation coefficients for the uncertainties of both methods. This appendix shows that HybridFlow consistently has lower correlation between uncertainty types than the NLL and BNLL methods, supporting our claim that the two uncertainties are distinct and disentangled with the model framework. The text has been updated in section 5.2 to reflect this.

---

> ### Author Response · Authors · 2025-08-08
> **Cont'd Response to Reviewer ijDQ**
>
> **Minor Points (in order)**
> - Thank you for this suggestion. As mentioned above, while deep ensembles are an important class of epistemic uncertainty estimators, this work does not aim to benchmark epistemic methods in isolation. Since HybridFlow is agnostic to the choice of epistemic predictor, we chose to focus the related work discussion on approaches that jointly model aleatoric and epistemic uncertainty. We now clarify this in section 4.1.
> - This is an excellent point. We have revised the manuscript to make it clear that the quantities we compute are estimates of aleatoric and epistemic uncertainty, not the ground truth values.
> - We have added a new appendix that provides precise mathematical descriptions of all the evaluation metrics used in our experiments, including the aleatoric and epistemic ECEs.
> - We apologize for the confusion. The deep ensemble in the ice sheet emulator experiments was a typo in the original table. This has been corrected in the revised version. The models are compared against a non-probabilistic LSTM (no UQ).

---

> > ### Comment · Reviewer_ijDQ · 2025-08-11
> > **Discussion**
> >
> > Thank you for your response. I still have a few issues with the work though:
> >
> > 1. Thank you for clarifying, I understand you are not using $y$ as input during inference, but I am now confused as to how meaningful the representation $z$ you obtain is. If you trained the flow to learn $p(y|x)$, then samples from the flow are obtained as $y=f(z|x)$, where $z$ is a standard Gaussian, sampled independently of $x$. If you then plug-in this generated sample $y$ to obtain $z = f^{-1}(y|x)$, how is this not just a Gaussian which is independent of $x$ by construction (i.e. by invertibility of the flow)? In other words, why is $z$ a meaningful representation of $x$ rather than just independent Gaussian noise?
> >
> > 2. You are not really addressing my concern about the inputs: if $z$ contains no additional information about $y$ than is already contained in $x$ (see point above), conditioning on $x$ alone or on both $x$ and $z$ would measure the same quantity: even if the estimates end up being different due to neural network training stochasticity, my concern is about how principled the approach is.
> >
> > 3. I disagree with your argument for not including comparisons against ensembles. I believe these comparisons should be included for the paper to be of interest to the UQ community.
> >
> > 4. Thank you for including the scatterplots, those do partially alleviate my concerns, but I would still like to clarify points 1 and 2 above.

---

> ### Author Response · Authors · 2025-08-18
> **Response to Reviewer ijDQ**
>
> **Comment 1**
> Thank you for raising this important point. We agree that our earlier description may not have made the role of $z$ sufficiently clear. In HybridFlow, $z$ is not simply an independent Gaussian noise variable; rather, it is an $x$-conditioned latent representation obtained through the invertible map of the conditional flow. Specifically, during training, the flow is optimized to model $p(y \mid x)$ via
>
> $y=f(z∣x)$.
>
> Although the base distribution for $z$ is Gaussian, the transformation $f(\cdot \mid x)$ depends explicitly on $x$. This means that the latent variable $z = f^{-1}(y \mid x)$ is not arbitrary noise, it encodes where a particular output $y$ lies within the $x$-conditioned distribution. Different values of $x$ induce different invertible mappings, so the resulting latent coordinates carry structured information about the conditional output space.
> At inference time, sampling $z$ and transforming via $f$ yields diverse plausible outputs consistent with the learned conditional distribution. Feeding $z$ alongside $x$ into the predictor allows the downstream model to exploit this structured latent representation, rather than treating uncertainty as unstructured noise. In practice, this provides the predictor with information about the variability captured by the flow and leads to improved calibration, as demonstrated in our ablation study (Appendix F). We will revise Section 3.2 of the manuscript to clarify the distinction between unconditional Gaussian noise and the $x$-dependent latent representation obtained through the conditional flow.
>
>
> **Comment 2**
> We thank the reviewer for this insightful comment. We agree that if the latent variable $z$ contained no additional information about $y$ beyond what is already in $x$, then conditioning on both would be redundant. However, in HybridFlow, $z$ is not an arbitrary latent variable, it is explicitly constructed by the conditional normalizing flow to encode the data-specific structure of the distribution of $p(y|x)$. This means that while $z$ is derived from $x$, it captures variability that is not directly accessible from $x$ alone. In practice, this enriches the feature space available to the predictor and allows more accurate and better calibrated estimates of uncertainty. Our results in both depth estimation and UCI regression show that this conditioning on both consistently improves predictive accuracy and calibration compared to conditioning on $x$ alone. Thus, while in principle redundancy is possible, in practice $z$ carries structured uncertainty information that is beneficial for uncertainty quantification. We have further clarified this point in the revision.
>
>
> **Comment 3**
> Thank you for voicing this concern. We have now included the comparison against the Deep Ensemble of heteroscedastic regression models as you suggested.

---

> > ### Comment · Reviewer_ijDQ · 2025-08-18
> >
> > Thank you for your reply. I am unfortunately still not convinced that your method is principled.
> >
> > 1. I still believe that the $z$ you are using is equivalent to Gaussian noise. To be more precise in notation, let's say you are given $x$ and you generate $\hat{y}$ by randomly sampling a Gaussian $\hat{z}$ and then setting $\hat{y}=f(\hat{z}|x)$. Then, the representation you compute, $z$, is defined as $z=f^{-1}(\hat{y}|x)$. My point is that, by construction, $f(\cdot|x)$ is invertible for every $x$, and you are using the same $x$ when computing $\hat{y}=f(\hat{z}|x)$ and when computing $z=f^{-1}(\hat{y}|x)$. In other words, you are computing $z=f^{-1}(\hat{y}|x) = f^{-1}(f(\hat{z}|x)|x) = \hat{z}$, so I don't see how the fact that the flow depends on $x$ makes it so that you are not just using Gaussian noise as an additional input.
> >
> > 2. I also still don't see how $z$ could have any additional information not present in $x$. Can you formally write down $z$ as a function which depends on something beyond $x$ and independent noise?
> >
> > 3. Thank you having run the additional experiments with deep ensembles. I am surprised at how poorly ensembles performed; in my experience they tend to improve performance over single models, which does not seem to be the case in Table 3. How many ensemble members did you use? Can you confirm that each ensemble member was trained in the exact same way as the single models from the "NLL" column?

---

> ### Author Response · Authors · 2025-08-19
> **Response to Reviewer ijDQ**
>
> **Comment 1:**
> Thank you for your persistence in understanding this work. You are correct that, due to invertibility, the latent $z$ is distributed as Gaussian noise. However, this does not mean that $z$ is “just noise” in the sense that it is uninformative. The use of $z$ enables us to target a specific, structured location within the conditional distribution $p(y|x)$. In practice, instances of $z$ can be thought of as different plausible outcomes of $y$, and conditioning the predictor on $z$ embeds knowledge of which region of the conditional distribution the point falls in. This usage is standard in existing literature of conditional normalizing flows; the conditional flows deliberately transform Gaussian noise distribution into a rich, $x$-dependent latent coordinate system that represents the conditional output distribution. For examples of this precedent in literature, here are a few examples:
> - **Hybrid Models with Deep and Invertible Features (Nalisnick et al., 2019)**: Showcases a Hybrid architecture with z as a direct input into a predictor.
> - **Learning Likelihoods with Conditional Normalizing Flows (Winkler et al., 2019)**: Conditioning on Gaussian latent variables is principled and is common practice.
> - **Normalizing Flows for Probabilistic Modeling and Inference (Papamakarios et al. 2021)**: Definitive survey of flows. Explicitly states that Gaussian base noise, though simple, becomes highly informative once transformed by conditional invertible maps.
> - **RealNVP (Dinh et al. 2017)** and **Glow (Kingma & Dhariwal 2018)**: Foundational generative flow models. Both start from Gaussian noise and show that invertible mappings turn this noise into richly structured samples that match complex data distributions.
> - **Noise Flow: Noise Modeling with Conditional Normalizing Flows (Abdelhamed et al. 2019)**: Treats noise itself as the signal of interest. Demonstrates that Gaussian latents in conditional flows are not “uninformative,” but capture structured noise properties that matter in practice.
> - **Conditional Density Estimation with Bayesian Normalizing Flows (Trippe & Turner, 2018)**: Highlights how conditioning on Gaussian latents provides Bayesian, uncertainty-aware density estimates.
>
> As stated in Section 3.1, $z$ is sampled from a Gaussian distribution, but that does not mean that $z$ is uninformative. However, if accepted, this can be clarified in the camera-ready version.
>
>
> **Comment 2:**
> We appreciate the reviewer’s concern and the chance to clarify. The role of the latent $z$ is not to introduce new information beyond $(x,y)$, but to provide a reparameterization of $y$ that makes conditional sampling and uncertainty estimation tractable. While $(x,z)$ and $(x,y)$ are equivalent at training time due to the bijection $z=f^{-1}(y|x)$, the latent coordinates allow us to factorize the conditional distribution into a structured component ($x$) and a latent component ($z$).
>
> At training time, $z=f^{-1}(y|x)$ is tightly coupled to $y$, so $I(z;y|x) = H(y|x)$ (Papamakarios et al., 2021, MacKay, 2003 section 8.1). This essentially means that $z$ captures the variability in $y$ once $x$ is known. This guarantees that $z$ encodes the same conditional information as $y$, rather than being unstructured noise.
>
> At inference time, we indeed use $z$, but the informativeness lies not in $z$ alone, but in the learned map $f(\cdot|x)$ that transforms $z$ into structured samples from $p(y|x)$. In other words, $z$ is a coordinate system for the conditional distribution. This is standard practice in conditional generative modeling (e.g., Glow, RealNVP, Nalisnick et al., Winkler et al., Papamakarios et al.), where the latent is not discarded but leveraged precisely for this role.
>
> Thus, $z$ is not “redundant Gaussian noise.” It is a functional reparameterization that converts conditional density estimation into tractable inference and enables calibrated uncertainty quantification. While this may seem like a technical distinction, it is the basis for why such approaches are widely adopted in conditional normalizing flows, variational autoencoders, and related frameworks. Our empirical results confirm that this representation is not only theoretically justified but also practically beneficial.
>
> **Comment 3:**
> Thank you for this comment. As we explicitly state in section 4.1, the ensemble contains 5 NLL models that were trained and tuned in the same way that the single-models from the NLL column were. We will further clarify this in the camera-ready version.

---

> ### Comment · Reviewer_ijDQ · 2025-08-19
>
> Thank you again for your continued attempts at clarifying my concerns. I maintain the concerns below.
>
> Re comment 1: You mentioned that you are using normalizing flows to obtain data representations as commonly done in the literature. I do not believe that to be the case at inference time. I understand that during training, when $y$ and $x$ are given, $z = f^{-1}(y|x)$ can be understood as a data representation, and that it is not independent of $x$ even if it follows a Gaussian distribution. Nonetheless, this would typically be understood as a representation of $y$, not a representation of $x$, so I disagree that you are using flows as is standard in the literature.
>
> However, the main issue arises at inference time. As I wrote in my previous comment, during inference, you obtain $\hat{z} = z$, so that by applying the flow to a generated sample, you simply revert it back to a Gaussian which is independent of $x$.
>
> Re comment 2: Again, I understand that $z$ is not independent of $x$ during training. The independence I am mentioning arises only during inference, and it is caused by not observing $y$ and using a generated sample instead. You claim "the informativeness lies in  $f(\cdot|x)$", but this depends only on $x$.

---

### Review · Reviewer_Edrs · 2025-07-28

**Summary Of Contributions:**

The authors propose an approach for estimating aleatoric and epistemic uncertainty, called HybridFlow.

HybridFlow entails separately training a conditional flow model to estimate $p(y|x)$, which at test-time is sampled to obtain an estimate of _aleatoric_ uncertainty.

The latent variable $z$ from the flow model is also fed as additional input to e.g. a deep ensemble or MC dropout model (which is trained separately in a second stage), which outputs a model prediction and an estimate of _epistemic_ uncertainty.

The approach is evaluated on the NYU Depth v2 dataset for dense depth estimation and on 12 datasets from the UCI regression benchmarks, being compared with heteroscedastic regression models. It's also applied to two ice sheet dynamics datasets.

**Audience:**

Yes

**Broader Impact Concerns:**

No concerns.

**Claims And Evidence:**

No

**Requested Changes:**

Overall, I think this is a quite interesting and solid paper, and I definitely think it could be relevant for the TMLR audience.

However, I think the current version requires some clarifications and tweaks, see "Questions/suggestions" above.

**Strengths And Weaknesses:**

Strengths:
- Uncertainty estimation for regression is somewhat understudied compared to classification, it's an interesting and important problem.
- The paper is well written overall, it contains basically not a single typo or similar issues. Figure 1 illustrates the proposed approach well.
- The proposed approach, to separately train a conditional flow model just for doing aleatoric uncertainty estimation, is quite interesting and makes sense overall.
- The proposed approach seems to perform well overall compared to heteroscedastic regression baselines.




Weaknesses:
- Section 4 is not ideally structured, and some experimental/implementation details are somewhat unclear.
- The simple baseline of creating a deep ensemble of Gaussian NLL models is not included in any of the experiments.
- The experimental results (especially Table 5) are somewhat difficult to overview and interpret (a lot of datasets and different metrics).




Questions/suggestions:
- Why do you never compare the uncertainty estimation quality to a regular deep ensembles model, I would expect that to outperform MC dropout? Should at least add this baseline to Table 6?
- It's not entirely clear to me how $x$ and $z$ are combined as input to the model? If $x$ is an image, is a feature vector first extracted, which is concatenated with $z$?
- Why is $z$ combined with $x$ as input to the model, is it obvious that this should help improve the performance? Have you done a direct ablation, comparing performance with/without $z$? In Section 5.2 you write _"This suggests that the latent feature representations learned by the NF contribute meaningful additional structure to the data, improving predictive performance beyond what is achievable with MSE optimization alone"_, but is $z$ really the only difference compared to that non-probabilistic MLP? No? Your model also adds MC dropout, which should affect the RMSE performance as well?
- In Table 1 and 2, I suppose that NLL and B-NLL also are implemened with MC dropout in order to estimate epistemic uncertainty? This could be made a bit more clear.
- You sample $\tilde{y} \sim p(y|x)$ in order to get an aleatoric uncertainty estimate, how many samples do you use?
- CMAF with 4 flow layers in Section 4.1, but 5 layers in 4.2? Any particular reason why you don't use the same number of layers?
- MC dropout with 30 samples in Section 4.1, but 50 samples in 4.2? Any particular reason why you don't use the same number of samples?
- The uncertainty estimation metrics are described in both Section 4.1 and 4.2? Better to do this just once before 4.1? Also, _"We evaluate aleatoric, epistemic, and total uncertainty with all of the aforementioned metrics. Total uncertainty, within the scope of this work, is the sum of the aleatoric and epistemic uncertainty, which is common in uncertainty quantification literature (Depeweg et al., 2018; Hüllermeier & Waegeman, 2021; Wimmer et al., 2023)"_ at the end of 4.2 is relevant for both 4.1 and 4.2, and could thus also be put before 4.1?
- In Section 4.2, the paragraph _"In addition to the heteroscedastic regression baselines, we also benchmark against CBDL (Caprio et al.,2024), a model-specific uncertainty framework [...] that is simpler to train and can be used with existing predictive models"_ does perhaps seem a bit superfluous? Or, could at least make more sense to describe this somewhere else, not in the experimental details for the UCI benchmarks here in Section 4.2?
- In Table 4, why are the training run-times sometimes so different for NLL and MLP? Should NLL not always be very similar, or a bit slower? Also, how come that HybirdFlow is almost 10x slower than MLP on "Superconductivity" but basically the same on "Yacht Hydrodynamics"?




Minor things:
- In Table 2, why not aleatoric, epistemic and total uncertainty also for PICP and CRPS? Just to save space?
- Start of Section 5.1: _"outperforming baseline models that use heteroscedastic regression methods and the CBDL model"_, but CBDL is not included here in 5.1?
- The Table 2 caption is closer to Table 1 than to Table 2, could perhaps tweak this a bit.
- Table 5 is too wide.
- Figure 1 is not referenced anywhere in the text?

---

> ### Author Response · Authors · 2025-08-08
> **Response to Reviewer Edrs**
>
> **Sentence 4 Structure**
> Thank you for this observation. In response, we have reorganized Section 4 to better separate methodological context from implementation and experimental details. Shared material (such as uncertainty metrics and architectural design choices) has been consolidated to reduce redundancy, and details that were previously embedded in Section 4.2, such as the discussion of CBDL, have been moved or removed to improve clarity and focus. The section now serves more clearly as a description of experimental design decisions and avoids revisiting concepts covered in the main methodology.
>
> **Added Deep Ensemble Baseline**
> We appreciate this suggestion. As noted in our response to your comment below, we use MC Dropout as a fixed epistemic uncertainty estimator across all models in order to isolate the contribution of the hybrid architecture. While Deep Ensembles may offer improved epistemic uncertainty in some cases, our focus is not on comparing epistemic mechanisms, but rather on how modeling aleatoric structure via a learned latent variable improves predictive and uncertainty performance. We now clarify this design decision more explicitly in Section 4.1. Additionally, we have included an Appendix section comparing MC Dropout vs a Deep Ensemble in the Ice Sheet Emulator case study.
>
> **Experimental Results (Table 5) Difficult to Interpret**
> This is an important suggestion, and we thank the reviewer for bringing this up. We have reduced the size of the table by only including total uncertainty metrics and moving the complete Table 5 to the Appendix. We’ve included the key takeaways and essential insights in the accompanying text, and we’ve designed the table structure to support interpretability (grouping metrics, consistent ordering, etc.). We believe that this showcases, in a robust way, the performance of our model on a variety of applications and tasks. That said, we are open to streamlining the presentation further in the camera-ready version if needed.

---

> ### Author Response · Authors · 2025-08-08
> **Cont'd Response to Reviewer Edrs**
>
> **Deep Ensembles vs MC Dropout**
> Thank you for this suggestion. We chose MC Dropout for its simplicity and to align with prior baselines such as Kendall & Gal. As noted in Section 4.1, while methods like Deep Ensembles may yield better-calibrated epistemic uncertainty, our goal is to isolate the effect of the hybrid architecture by holding the epistemic method constant. We expect the relative model comparisons to remain similar even with Deep Ensembles, as the epistemic mechanism has limited impact on predictive performance. We now clarify this point more explicitly in Section 4.1 in response to this comment. Additionally, we have included an Appendix section comparing MC Dropout vs a Deep Ensemble in the Ice Sheet Emulator case study.
>
> **Inputs x and z into HybridFlow**
> Thank you for pointing this out. We now clarify in Section 3.2 that when x is high-dimensional (e.g., images), a feature extractor may be used to reduce the dimensionality of the input to the normalizing flow. However, the full input x is still provided to the probabilistic predictor alongside the latent variable z, ensuring that no information is lost at prediction time.
>
>
> **Why to combine x and z...**
> Thank you for the thoughtful comment. Per your suggestion, we completed an ablation study and included it in Appendix F. This section compares HybridFlow models using both x and z, only x, or only z as inputs, along with a non-probabilistic MLP baseline. The results show that combining x and z typically improves RMSE, particularly on complex datasets. We've updated Section 5.2 to refer to the appendix.
>
> **Table 1 and 2 MC Dropout Clarification**
> Yes, thank you for the question. This point has been further clarified in section 4.1.
>
> **Aleatoric Samples**
> The text has been updated to include this information in both 4.1 and 4.2.
>
> **Flow Layers and MC Dropout Samples.**
> Thank you for pointing this out. We have updated the text to reflect the most updated version of the architecture. The discrepancy is a remnant from a previous version, which has now been updated and reflected in the manuscript.
>
> **UQ Explanation Redundancy**
> The redundancy has been removed and included only in 4.1, including the definition of total uncertainty.
>
> **CBDL Explanation Location**
> Thank you for this suggestion. This paragraph has now been largely removed, with a few key points integrated into the Related Work section.
>
> **Runtime Considerations & Complexity**
> Thank you for pointing this out. We agree that, in principle, the NLL-based models should have similar training time to the MLP baselines, since they share the same architecture with a small addition (a variance head) and a different loss function. In practice, however, we observed some variability in runtime due to two main factors: 1) early stopping behavior, and 2) differences in convergence stability. The MLP models often converge slightly faster under MSE loss, while the NLL and especially BNLL models can require more epochs due to variance instability or optimization sensitivity. This is particularly noticeable in small datasets, where differences of just a few epochs can lead to visible shifts in runtime due to the small absolute scale.
>
> For HybridFlow, the variation in relative training time across datasets is primarily due to differences in dataset size and dimensionality. On larger, high-dimensional datasets like Superconductivity (21,000+ samples, 81 input features), the CMAF does increase computation in a noticeable way. Conversely, on small, low-dimensional datasets like Yacht Hydrodynamics, the NF component is lightweight and converges very quickly, making the added cost of HybridFlow negligible.
>
> Overall, while there are some dataset-dependent variations, the key point is that (1) the NLL models and MLP baselines are indeed similar in most cases, and (2) HybridFlow generally incurs a modest increase in runtime relative to the added benefit of improved uncertainty quantification and accuracy. We believe the tradeoff is reasonable and reflected in the discussion in the main text and Appendix.
>
>
>
> **Minor Suggestions (in order)**
> - Thank you for the question, we don’t report separate aleatoric and epistemic values for PICP and CRPS because these metrics evaluate properties of the full predictive distribution (e.g., interval coverage and overall distributional accuracy). Since aleatoric and epistemic uncertainty are modeled separately and then combined to form the final predictive distribution, PICP and CRPS only make sense when computed on the total uncertainty.
> - Thank you for finding this. We have removed the reference to CBDL here.
> - The spacing has been updated to better separate the tables.
> - The table has now been moved to the appendix
> - A reference to figure 1 has now been included in Section 3.2.

---

> ### Comment · Reviewer_Edrs · 2025-08-11
>
> Thank you for the response.
>
> I have read the two other reviews and all rebuttals.
>
> The other reviews are quite similar to mine overall, they don't change my impression of the paper in any obvious way.
>
> The authors have provided a solid rebuttal overall, responding to most questions/concerns, adding clarifying experiments to the appendix, tweaking the main paper etc.
>
> However, two things remain unclear to me:
>
>
>
> ***
> ***
> ***
> ***
> ### ***Q1***:
>
> _"Inputs x and z into HybridFlow: Thank you for pointing this out [...] However, the full input x is still provided to the probabilistic predictor alongside the latent variable z, ensuring that no information is lost at prediction time"_
>
> How exactly is this done? Are $x$ and $z$ concatenated and then input to the model? But what if $x$ is an image? How are then the image $x$ and the vector(?) $z$ combined?
>
>
>
>
>
> ***
> ***
> ***
> ***
> ### ***Q2***:
>
> My original weakness _"The simple baseline of creating a deep ensemble of Gaussian NLL models is not included in any of the experiments"_ has still not been addressed.
>
> Reviewer ijDQ also specifically asked for this (_"I would like to see a comparison against an ensemble of heteroscedastic regression models, where aleatoric uncertainty is estimated as the average (over ensemble members) predicted variance, and where epistemic uncertainty is estimated as the variance (over ensemble members) of the mean prediction"_).
>
> Creating an ensemble of Gaussian NLL models is basically the simplest possible uncertainty estimation baseline for regression problems. I would like to see the proposed HybridFlow being compared with this simple baseline, in order to demonstrate that HybridFlow actually provides practical benefit on real-world datasets.
>
>
>
>
> ***
> ***
> ***
> ***
> Minor things:
> - Section 3.4, "paired with an feature extractor or autoencoder" --> "paired with a feature extractor or autoencoder".
> - Section 3.4, "that scales approximately linearly number of flow layers" --> "that scales approximately linearly with the number of flow layers"?

---

> > ### Author Response · Authors · 2025-08-18
> > **Response to Reviewer Edrs**
> >
> > **Q1**
> > Thank you for emphasizing the clarity of this part. We have now clarified further in section 3.2.
> >
> > **Q2**
> > Thank you for this suggestion. We have now included the comparison in the main text to showcase the benefit of HybridFlow.

---

> ### Comment · Reviewer_Edrs · 2025-08-24
>
> Thank you for providing additional details and results.
>
> Q1 is now clear, thank you.
>
> For Q2, given that "Deep Ensemble" should be an ensemble of 5 models trained in exactly the same way as "NLL", I find it strange that Deep Ensemble achieves worse MSE in Table 1, and worse RMSE for 11/12 datasets in Table 3. This doesn't really make sense to me. Also, "Deep Ensemble" results are missing in Table 5.

---

> ### Comment · Reviewer_Edrs · 2025-08-28
>
> After further discussion with the other reviewers, I also have questions related to your discussion with reviewer ijDQ.
>
> How do you actually obtain $z$ at test-time?
>
> Do you sample $z$ from a standard Gaussian, use this to generate a sample $y$ from the trained flow model, and then map this generated sample $y$ back to $z$, and use this $z$ as input together with $x$? If so, I don't really see how this could add any information compared to just inputting $x$?
>
> Or, is $x$ mapped directly to $z$? (_"1. Data Transformation: Input data x is transformed into a latent representation z via a CMAF- based NF, conditioned on x"_ (start of Section 3.4)) I.e., similar to page 4 in https://arxiv.org/pdf/1912.00042 where they have $p(z|x)$? In that case, this $z$ would be some kind of representation of the predictive distribution $p(y|x)$ of the trained flow model, for this given $x$. That it might help a bit to give this $z$ as additional input together with $x$, sort of makes sense to me.
>
> Hopefully you can help me understand this properly.

---

> ### Author Response · Authors · 2025-09-02
> **Response to Revewer Edrs**
>
> Thank you for your thoughtful feedback and for providing the opportunity to clarify these points. Your comments and deliberation with other reviewers has been invaluable in helping improve the clarity and rigor of the paper.
>
> **Deep Ensemble Performance**
> The question of why a Deep Ensemble (DE) might underperform a single NLL model on prediction metrics highlights a key aspect of the DE methodology. You are correct that, in many cases, an ensemble performs at least as well as its individual members. However, this relies on the assumption that all members have converged to a meaningful and well-placed local minimum. Current literature has highlighted (Seitzer et al., "On the Pitfalls of Heteroscedastic Uncertainty Estimation with Probabilistic Neural Networks"; Stirn et al., "Faithful Heteroscedastic Regression with Neural Networks"; Wong-Toi et al., "Understanding Pathologies of Deep Heteroskedastic Regression") training models with a heteroscedastic NLL loss is subject to significant training instabilities. These models can easily get stuck in poor local minima or fail to converge meaningfully depending on model initializations and optimization trajectories.
>
> The Deep Ensemble framework, as proposed by Lakshminarayanan et al., is designed to robustly capture epistemic uncertainty by leveraging the diversity of outcomes from multiple training runs. The methodology explicitly discourages hyper-optimization or cherry-picking individual models within the ensemble, but rather to rely on the natural variability between models based on initializations, bootstrapping, etc. In this revision, we follow this methodology exactly, meaning that we did not restart or discard models. Consequently, the performance of the five-model ensemble reflects and average over all members, including any that may have settled in a suboptimal part of the loss landscape. This implementation is consistent with a true DE, rather than creating an "ensemble of individually optimized models". For completeness, below is a table of results comparing the single NLL model, the true DE model, and the Ensemble of NLL models results. The manuscript can be updated based on reviewer comments to reflect the Ensemble of NLL models, but currently the DE remains and a clarification has been added to the discussion.
>
> | Method | RMSE | AbsRel     | $\delta_1$ | $\delta_2$ | $\delta_3$ |
> | :---        |    :----:   |          :---: | :---: | :---: | ---: |
> | NLL (single)  |  0.129        | 0.083         |  0.947        |  0.987	| 0.995		|
> | Deep Ensemble |  0.166	| 0.100        	|  0.927        |  0.985	| 0.994		|
> | Ensemble of NLL models  |  0.116 | 0.074       	|  0.952        |  0.990        | 0.995         |
> | HybridFlow	|  0.058	| 0.041        	|  0.989        |  0.998	| 0.999		|
>
>
>
> **Method for Obtaining $z$ at Test-Time**
> Thank you for raising this question and referencing the Winkler et al. paper, as it highlights a point of confusion in the notation that can be addressed.
>
> Your second hypothesis is correct. To clarify, $x$ is indeed mapped directly to $z$ by using the learned base distribution. Notation within Normalizing Flow (NF) literature can differ slightly based on which direction you deem to be the "forward" pass (which can make things confusing) but our equation 3 matches with the Winkler et al. equation 7. As you say, $z$ serves as a feature vector representing the predictive distribution learned by the flow model. Our notation $f^{-1}(y|x)$ can be seen as the equivalent to $p(z|x)$ in a sense, as our notation emphasizes the usage of the flow to model this distribution. We also emphasize more clearly in the text that in conditional settings, the base distribution is a conditional Gaussian distribution, and that at test time we obtain a latent $z$ from the expectation of the learned distribution $p(z|x)$.
>
> In the previous version of our manuscript, we began with notation for unconditional flows and then directly specified conditional flows, which may have been causing confusion as to the information contained within z. We have updated the notation and text within section 3.1 to clarify and focus on conditional notation (particularly with the base distribution) in section 3.1 and added further details in 4.1.

---

> > ### Comment · Reviewer_Edrs · 2025-09-03
> >
> > Thank you, this clarifies things.
> >
> > I'm still not entirely sure how different this proposed approach would be in practice compared to straightforward ensembling, if this actually would provide significantly more high-quality uncertainty estimates in real-world applications. But, the paper is still solid and quite interesting, and I definitely think it could be relevant for the TMLR audience.
> >
> > Thus, I will recommend accept.

---

### Review · Reviewer_H6s6 · 2025-07-28

**Summary Of Contributions:**

The authors propose HybridFlow, an approach that models aleatoric uncertainty using a Conditional Masked Autoregressive Flow and epistemic uncertainty using a predictor such as deep ensembles. They argue that employing a separate normalizing flow to capture aleatoric uncertainty helps preserve generalization performance and does not interfere with the modeling of epistemic uncertainty.
Predictions are generated by the predictor model using its expected value. Epistemic uncertainty is estimated as the variance across the predictor’s outputs, while aleatoric uncertainty is derived from the sample variance of the predictive distribution produced by the normalizing flow.

**Audience:**

Yes

**Claims And Evidence:**

Yes

**Requested Changes:**

See above.

**Strengths And Weaknesses:**

**Strengths**:
- The paper is well-written and easy to follow.
- The core idea of decoupling aleatoric and epistemic uncertainty using separate models is both intuitive and valuable. The authors motivate this effectively with clear examples.
- The proposed method is simple and accessible, making HybridFlow appealing to both researchers and practitioners.

**Weaknesses**:
- **Repetitions, conciseness, and representation**:  The paper would benefit significantly from a more concise presentation. There are frequent repetitions, especially regarding the modularity of the approach, which could be removed without loss of clarity. Additionally, the abstract and introduction describe HybridFlow too vaguely. It should be clearly stated which type of normalizing flow is used, and that, in principle, any predictor (e.g., deep ensembles) can be integrated. From a layout perspective, I recommend avoiding sections that consist solely of bullet points. The workflow summary should instead be integrated into the training protocol at the beginning of Section 3.5.

- **Computational considerations**: Section 3.5 mentions the use of an autoencoder to address the high dimensionality of the output space. However, the computational cost of the full method remains unclear. How does the complexity of the model scale with the dataset's complexity? Is a more complex normalizing flow necessary for more challenging datasets? This point is important for practical applications and should be at least discussed. Ideally, the authors could include additional experiments (e.g., in the appendix) to support this.

- **Choice of normalizing flow**: As someone not deeply familiar with normalizing flows, I wonder why the Conditional Masked Autoregressive Flow (C-MAF) was chosen. Is it particularly well-suited for HybridFlow, or could other flows be used as well? Some justification for this design choice would be helpful. I would have expected a comparison with at least one alternative normalizing flow to validate this aspect.

- **Novelty and related work**: The novelty of the proposed method is somewhat difficult to assess without a stronger contextualization in related literature. The current related work section is rather general. A more focused discussion of prior work using normalizing flows in a similar way would help clarify the paper’s contribution. If no such work exists, the authors should clearly state that they are the first to model aleatoric uncertainty with a normalizing flow.

- **Minor remark**: Ensure that all tables use a consistent font size and span the full `\linewidth`.

---

> ### Author Response · Authors · 2025-08-08
> **Response to Reviewer H6s6**
>
> **Repetitions, conciseness, and representation**
> Thank you for pointing this out. The manuscript has been extensively reviewed to cut down on the frequent repetitions you mention. The abstract has been updated, describing the specifics of the CMAF and the flexibility of the epistemic module. The summary of the workflow has also been incorporated into section 3.5 as recommended.
>
> **Computational considerations**
> Thank you for emphasizing this crucial aspect of the paper. We have added text in Section 3.4  (last paragraph) that discusses how the size of the flow impacts computational complexity. We ran an ablation study (included in Appendix A) that showcases the training time on several UCI benchmarks with varying flow depths to show how runtimes and accuracy changes with deeper architectures.
>
> **Choice of normalizing flow**
> Thank you for raising this point. We chose Conditional Masked Autoregressive Flows (CMAF) due to their strong performance on conditional density estimation tasks, efficient likelihood computation, and stable training dynamics. That said, HybridFlow is agnostic to the specific flow architecture: other conditional flows (e.g., RealNVP, Neural Spline Flows, c-GLOW) can be used with minimal changes. We have clarified this in Section 3.1.
>
> **Novelty and related work**
> This is an insightful suggestion. The related work section has been significantly revamped, establishing more context on using NFs for uncertainty quantification and how they are used. We present a more focused discussion on how HybridFlow fits in with previous works, and explicitly state the contribution in Section 2 (specifically the paragraph starting with “These limitations…”).

---

### Author Response · Authors · 2025-08-08
**Author Response to Reviewers**

We thank all reviewers for their constructive and detailed feedback, which has greatly improved the clarity, rigor, and presentation of this work. In response, we have substantially revised the manuscript to reduce redundancy, strengthen the abstract and related work, clarify methodological details, and reorganize the experimental section for improved readability. We have also added new analyses, ablations, and runtime studies to address specific concerns, and believe these changes comprehensively address the issues raised. Please see the revised version of the manuscript for the changes.

---

### Decision · Action_Editor_C8Z1 · 2025-09-16

**Recommendation:** Accept as is

**Audience:**

Yes

**Audience Explanation:**

All three reviewers agree that the paper is of interest to parts of the TMLR readership.
I agree with them.

**Claims And Evidence:**

Yes

**Claims Explanation:**

After a long rebuttal discussion and several changes to the paper, two out of three reviewers
are convinced by the provided evidence and recommend acceptance.
Some doubt remains about the suboptimal ensemble performance, and one reviewer remains doubtful
about the uncertainty decomposition interpretation.

Given the latest updates and the strong empirical performance, I agree with the majority
that the strengths of the paper outweigh its weaknesses.